# MIND: Modality-Informed Knowledge Distillation Framework for Multimodal Clinical Prediction Tasks

**Alejandro Guerra-Manzanares**  *alejandro.guerra@nyu.edu*
*Engineering Division*
*New York University Abu Dhabi*

**Farah E. Shamout**  *farah.shamout@nyu.edu*
*Engineering Division*
*New York University Abu Dhabi*

**Reviewed on OpenReview:** *https://openreview.net/forum?id=BhOJreYmur*

## Abstract

Multimodal fusion leverages information across modalities to learn better feature representations with the goal of improving performance in fusion-based tasks. However, multimodal datasets, especially in medical settings, are typically smaller than their unimodal counterparts, which can impede the performance of multimodal models. Additionally, the increase in the number of modalities is often associated with an overall increase in the size of the multimodal network, which may be undesirable in medical use cases. Utilizing smaller unimodal encoders may lead to sub-optimal performance, particularly when dealing with high-dimensional clinical data. In this paper, we propose the Modality-INformed knowledge Distillation (MIND) framework, a multimodal model compression approach based on knowledge distillation that transfers knowledge from ensembles of pre-trained deep neural networks of varying sizes into a smaller multimodal student. The teacher models consist of unimodal networks, allowing the student to learn from diverse representations. MIND employs multi-head joint fusion models, as opposed to single-head models, enabling the utilization of unimodal encoders in the case of unimodal samples without requiring imputation or masking of absent modalities. As a result, MIND generates an optimized multimodal model, enhancing both multimodal and unimodal representations. It can also be leveraged to balance multimodal learning during training. We evaluate MIND on binary classification and multilabel clinical prediction tasks using clinical time series data and chest X-ray images extracted from publicly available datasets. Additionally, we assess the generalizability of the MIND framework on three non-medical multimodal multiclass benchmark datasets. The experimental results demonstrate that MIND enhances the performance of the smaller multimodal network across all five tasks, as well as various fusion methods and multimodal network architectures, compared to several state-of-the-art baselines.

## 1 Introduction

Healthcare decision-making, like most human-based active reasoning (Smith & Gasser, 2005), is inherently multimodal. This means that clinical decisions are typically informed by multiple sources of information or modalities, such as diagnostic imaging, clinical time-series data, or medical history, combined with clinical experience and expertise (Bate et al., 2012; Pauker & Kassirer, 1980; Pines et al., 2023). As a result, recent research has demonstrated the potential of multimodal learning in modeling various clinical prediction tasks (Huang et al., 2020), particularly in information fusion to enhance downstream classification performance compared to relying solely on a unimodal network (Hayat et al., 2022; Yang et al., 2022; Wang et al., 2020a). Note that the term unimodal model refers to a modality-specific neural network, i.e., a model that processes

samples consisting of a single data modality (e.g., image), while the term multimodal model refers to a network that processes samples consisting of multiple data modalities.

Despite recent advancements, the nature of the learning process in multimodal clinical prediction tasks remains unclear. Most research primarily focuses on achieving optimal fusion performance (Huang et al., 2020), often overlooking additional challenges of multimodal training, such as optimizing modality utilization and modeling cross-modal interactions for effective training (Huang et al., 2022b; Wang et al., 2020b; Wu et al., 2022b). For instance, recent findings indicate that multimodal deep neural networks are prone to overfitting due to their larger capacity (Wang et al., 2020b). They tend to rely primarily on one of the available input modalities, specifically the one that is fastest to learn from (Wu et al., 2022b). Since different modalities may generalize and overfit at different rates, naive joint training often leads to sub-optimal results (Wang et al., 2020b).

The methodological challenges of multimodal learning are coupled with other challenges when applied in healthcare. This includes the limited availability of large-scale multimodal clinical datasets, as well as the sparse and heterogeneous nature of the input modalities, such as when fusing medical images with clinical time-series data. The increase in the number and heterogeneity of modalities entails the use of modality-specific encoders, such as convolutional neural networks for images and recurrent neural networks for time-series data (Hayat et al., 2022). This requirement increases the overall size of the multimodal network, which may hinder deployment in clinically relevant use cases such as privacy-preserving training (Baruch et al., 2022), secure inference (Lou et al., 2021), and applications on resource-constrained devices (Zhao et al., 2018).

Our objective in this work is to improve the compression of multimodal networks, both in terms of size and predictive performance when dealing with small multimodal datasets. We study this within the context of joint fusion, where two input data modalities have their encoded representations aggregated before applying a fusion layer. To achieve this objective, we propose a training framework based on knowledge distillation that incurs minimal modifications to the joint fusion architecture and learning objective, while significantly enhancing predictive performance and reducing overall size. We refer to this training framework as Modality-INformed knowledge Distillation (MIND). Specifically, we make the following contributions:

- We propose integrating modality-specific supervision signals into the joint fusion architecture. This approach enhances the representations of unimodal encoders by reducing the influence of the global model, allowing for a stronger focus on each unimodal encoder. As a result, this leads to enhanced performance in multimodal fusion.

- We propose pre-training unimodal teachers to distill knowledge into modality-specific signals, compressing the knowledge from an ensemble of larger unimodal models and enhancing the representations learned by the modality encoders of the student model. Our results demonstrate that this approach significantly improves fusion performance while utilizing a more compact student network.

- We introduce additional weighting hyper-parameters to emphasize modality-specific learning. Our empirical results demonstrate that this approach improves both unimodal and fusion (multimodal) performance, while also helping to balance modality learning during multimodal training.

## 2 Background

**Knowledge distillation.** Knowledge distillation (KD) (Hinton et al., 2015), also referred to as model compression (Buciluă et al., 2006) or teacher-student networks (Gou et al., 2021), allows transferring the generalization capability of a large model into a typically smaller model. The most common technique is response-based distillation, which leverages the output class probabilities of the larger model as soft targets for training the smaller model (Hinton et al., 2015). In the offline knowledge distillation setting, the aim of the student network is to mimic the performance of the teacher network by approaching the softened responses of the pre-trained teacher network. The main rationale is that the function learned by a large model can be approximated by a shallower model, which is computationally less expensive to train (Gou et al., 2021). Although increased network depth can improve learning, it is not always necessary or desirable

(Ba & Caruana, 2014). Other approaches involve approximating the features and layers of models, or the relationships between instances, in online or self-distillation settings (Gou et al., 2021).

KD aims to transfer the knowledge and generalization capabilities of a large model to a smaller model (Hinton et al., 2015), typically using student-teacher networks (Gou et al., 2021). In a multiclass classification problem, the student network mimics the predictions of the teacher model during training (Gou et al., 2021) by minimizing:

$$\mathcal{L} = \underbrace{\mathcal{L}_S(y, p(z_s, \tau = 1))}_{\text{supervised learning}} + \underbrace{\mathcal{L}_{KD}(p(z_t, \tau), p(z_s, \tau))}_{\text{knowledge distillation}} \tag{1}$$

where $\mathcal{L}_S$ is the supervised learning loss applied to the student network (Cross-Entropy (CE) loss), $z_s$ are the logits of the student, $\tau$ is a temperature parameter that regulates the importance of each target class, and $p$ is the output probability obtained after applying softmax activation. $\mathcal{L}_{KD}$ is the KD loss, which quantifies the divergence between the teacher ($z_t$) and student outputs ($z_s$), either using the Kullback-Leibler divergence loss (KL) or CE. In a multiclass setting, when $\tau = 1$, the supervised loss directly yields $\hat{y}$. In contrast, when $\tau > 1$, KD generates the soft targets for the distillation loss.

**Multimodal learning & knowledge distillation.** Recent work leveraged KD for a variety of multimodal learning tasks, such as for generating a multimodal student from a unimodal teacher (Xue et al., 2021), addressing the limitations of CLIP (Dai et al., 2022), multimodal generation via cross-modal vision-language KD (Radford et al., 2021), and optimized action recognition using multimodal sensors via wavelet KD (Quan et al., 2023). Other studies focused on handling missing modalities and improving computational efficiency by shifting from multimodal to unimodal attention (Agarwal et al., 2021), or improving unlabeled image selection from a pre-trained vision-language model (Zhang et al., 2022). The findings of previous work highlight the versatility and effectiveness of KD in improving the performance of various multimodal learning tasks. In our work, we focus on KD in the context of multimodal fusion networks, leveraging it to compress knowledge from an ensemble of teachers.

**Knowledge distillation in healthcare**. KD has been utilized to compress deep neural networks for a wide range of clinical prediction tasks. For unimodal tasks, several studies explored KD for federated learning (Chen et al., 2022) and multi-institution collaboration (Huang et al., 2023), medical text classification (Huang et al., 2022a), electronic health record mining (Du & Hu, 2020), and multiclass classification for medical images (Yang et al., 2022; Ho & Gwak, 2020; Soni et al., 2019). Wang et al. (2020a) and Dou et al. (2020) explored KD in the context of multimodal imaging data for multiclass classification, as well as for multiclass cardiac structure and multi-organ segmentation, respectively. In another study (Dou et al., 2020), the authors proposed to learn shared representations among imaging modalities to distill knowledge for modality-specific segmentation, while Wang et al. (2020a) distilled the knowledge of a unimodal teacher into a multimodal student. Another work (Hu et al., 2020) investigated distilling knowledge from a multimodal teacher to a unimodal student network for clinical image segmentation. In general, the use of KD for multimodal clinical prediction tasks, and for multilabel classification in particular —- a relevant problem in healthcare applications where multiple labels may be simultaneously present in a sample — has been under-explored. In contrast, our work investigates the application of KD for model compression of multimodal networks for heterogeneous medical modalities, focusing on binary and multilabel classification tasks.

**Training of multimodal networks**. Joint training of multimodal fusion networks is a challenging task due to the tendency of these networks to overfit one of the input modalities (Wang et al., 2020b). As shown in recent work (Huang et al., 2022b), the different input modalities compete against each other during joint training, leading to sub-optimal joint training where only a subset of the input modalities is learned efficiently, while the rest remain unexplored and contribute minimally (Wu et al., 2022b). Several techniques have been proposed to generate a more balanced training approach for multimodal learning (Huang et al., 2022a; Wu et al., 2022b; Wang et al., 2020b; Peng et al., 2022), utilizing metrics to assess the degree of modality overfitting (Wang et al., 2020b; Wu et al., 2022b). In this work, we leverage the conditional utilization rate (Wu et al., 2022b) to demonstrate that, as a by-product of our proposed framework, the weights of the distillation loss can be used to emphasize learning from specific modalities. This approach improves

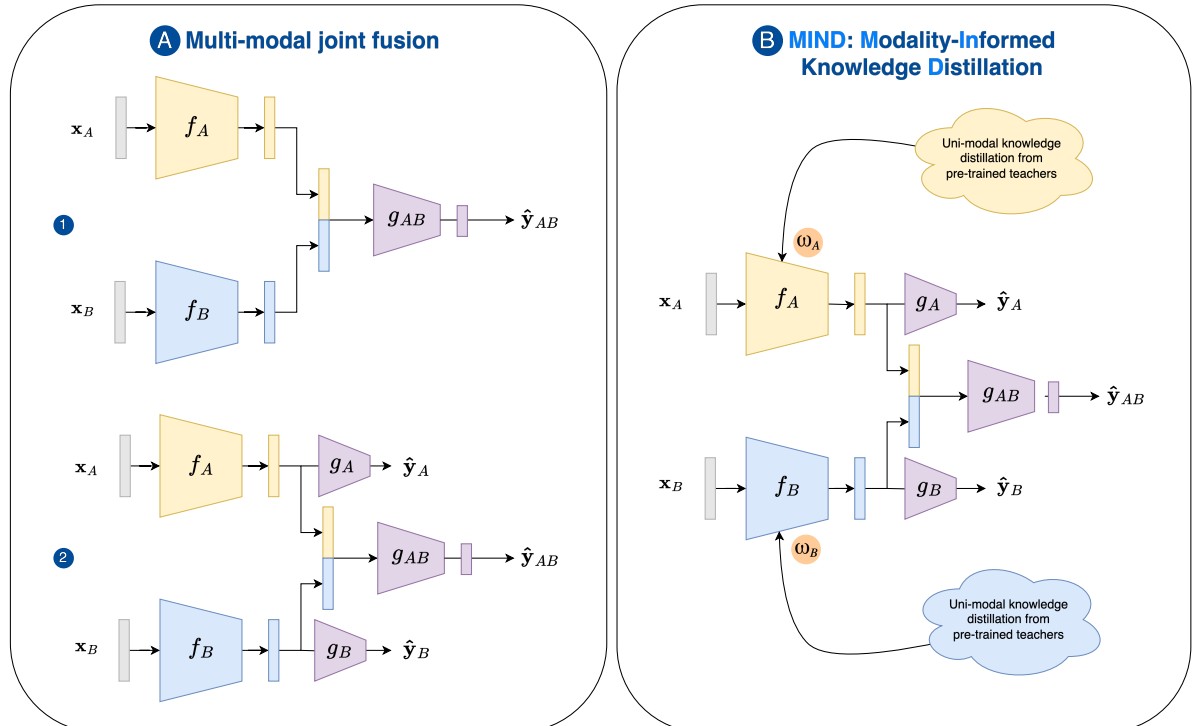

Figure 1: **Architecture of multimodal fusion network.** A.1 Architecture for multimodal joint fusion, as in recent work (Hayat et al., 2022). A.2 Modified base architecture that incorporates modality-specific classification heads with the loss shown in Equation 2. B. Architecture of the MIND framework, featuring the loss shown in Equation 5, which integrates unimodal pre-trained teachers for KD into the unimodal encoders. It also includes modality weighting hyper-parameters $(\omega_A, \omega_B)$ to emphasize encoder representation learning and improve modality utilization during multimodal training.

modality utilization and leads to a more balanced multimodal training process, enhancing both multimodal and unimodal predictions of the model trained via the MIND framework.

## 3   Methods

**Preliminaries.** In a simple multimodal fusion task, we assume the presence of two input modalities represented by $\mathbf{x}_A$ and $\mathbf{x}_B$. The goal of the multimodal fusion task is to jointly learn from both modalities to predict a set of ground-truth labels denoted by $\mathbf{y}$. Due to heterogeneity, each modality is first processed by modality-specific encoders, such that $\mathbf{z}_A = f_A(\mathbf{x}_A)$ and $\mathbf{z}_B = f_B(\mathbf{x}_B)$, where $\mathbf{z}_A$ and $\mathbf{z}_B$ are latent feature representations. These latent representations are concatenated and processed by a fusion classification layer, such that $\hat{\mathbf{y}}_{AB} = g_{AB}(\mathbf{z}_A, \mathbf{z}_B)$. We then apply the binary cross-entropy (BCE) loss, denoted by $\mathcal{L}_{S_{AB}}(\mathbf{y}, \hat{\mathbf{y}}_{AB})$, and optimize $f_A$, $f_B$, and $g_{AB}$ jointly, as shown in Figure 1 A.1.

**Multimodal loss.** We incorporate two additional loss terms by introducing classification modules for each modality, such that $\hat{\mathbf{y}}_A = g_A(\mathbf{z}_A)$ and $\hat{\mathbf{y}}_B = g_B(\mathbf{z}_B)$. Hence, the overall supervised learning loss becomes:

$$\mathcal{L}_S = \mathcal{L}_{S_{AB}}(\mathbf{y}, \hat{\mathbf{y}}_{fusion}) + \mathcal{L}_{S_A}(\mathbf{y}, \hat{\mathbf{y}}_A) + \mathcal{L}_{S_B}(\mathbf{y}, \hat{\mathbf{y}}_B) \tag{2}$$

This modification allows for the independent use of unimodal classification heads in the event of unimodal samples at inference time, as illustrated in Figure 1 A.2. Note that the term unimodal sample refers a data sample that contains only one modality (e.g., A or B), whereas a multimodal sample involves the presence of more than one modality (e.g., A and B).

**Unimodal teachers.** We use KD to compress the knowledge of an ensemble of unimodal teacher models into a single, typically smaller student model. KD was originally conceived for multiclass classification problems, where different values of $\tau$ are used to generate the soft targets. In our work, we propose to use KD for a wider range of predictive tasks including binary and multilabel classification tasks. For multilabel classification, we modify the $\mathcal{L}_{KD}$ term in Equation 1 to:

$$\mathcal{L}_{\mathrm{KD}}(\hat{\mathbf{y}}^s, \hat{\mathbf{y}}^t) = \frac{1}{N} \sum_{i=1}^{L} BCE(\hat{y}^{s,i}, \hat{y}^{t,i}), \tag{3}$$

where $L$ is the set of possible labels in a multilabel classification problem, $N$ is the sample size, and $s$ and $t$ denote the student and teacher networks, respectively.

Given an ensemble $T$ of $K$ teachers, we define a new average ensemble prediction for an $i$-th label (omitting $i$ without loss of generality):

$$\hat{y}^T = \frac{1}{K} \sum_{k=1}^{K} \hat{y}^{t_k}, \tag{4}$$

such that the Ensemble Knowledge Distillation (EKD) loss is now denoted as $\mathcal{L}_{EKD}(\hat{\mathbf{y}}^s, \hat{\mathbf{y}}^T)$.

In a comprehensive multimodal setting, the goal of the classification model is to accurately classify both multimodal and unimodal samples, which is particularly relevant in healthcare applications where some modalities may be absent in a sample. To improve the representations of multimodal and unimodal encoders, we introduce $\mathcal{L}_{EKD^U}$, which consists of an ensemble of unimodal teachers trained with sub-components of Equation 2 for each modality, namely $\mathcal{L}_{S_A}$ and $\mathcal{L}_{S_B}$. Hence, we introduce a new learning objective term for each input modality to distill knowledge from unimodal teachers to the modality encoders of a smaller multimodal student. In the case of two modalities (A and B), two terms are introduced:

$$\underbrace{\mathcal{L}_{EKD_A^U}}_{\substack{\text{unimodal ensemble} \\ \text{knowledge distillation} \\ \text{for modality A}}} , \quad \underbrace{\mathcal{L}_{EKD_B^U}}_{\substack{\text{unimodal ensemble} \\ \text{knowledge distillation} \\ \text{for modality B}}} \tag{5}$$

We note that goal of the multimodal student network is to learn the distribution $P(Y|X_A, X_B)$. To enhance encoder representations and improve the learning of the multimodal distribution, we introduce the unimodal EKD components, consisting of $\mathcal{L}_{EKD_A^U}$ and $\mathcal{L}_{EKD_B^U}$, which represent the distributions $P(Y|X_A)$ and $P(Y|X_B)$, respectively, as learned by the unimodal teachers.

**Balancing multimodal training.** We introduce weighting hyper-parameters $\omega_A$ and $\omega_B$ to adjust the focus on the distillation components. This enhances encoder representation and modality learning, resulting in the overall loss defined as follows:

$$\mathcal{L}_{\mathrm{MIND}} = \underbrace{\mathcal{L}_{S_{AB}} + \mathcal{L}_{S_A} + \mathcal{L}_{S_B}}_{\text{supervision signals}} + \underbrace{\omega_A \times \mathcal{L}_{EKD_A^U} + \omega_B \times \mathcal{L}_{EKD_B^U}}_{\substack{\text{weighted unimodal ensemble} \\ \text{knowledge distillation}}} \tag{6}$$

A visual representation of the proposed MIND framework is provided in Figure 1 B. In particular, setting $\omega_A, \omega_B > 1$ minimizes both loss components (supervision signals and EKD), emphasizing distillation for unimodal encoder learning. Specifically, $\omega_A, \omega_B \gg 1$ prioritize knowledge distillation, while $\omega_A, \omega_B = 1$ treat all loss components equally. When $\omega_A \gg \omega_B$, the learning focuses on modality A, whereas $\omega_A \ll \omega_B$ prioritizes modality B. This independent weighting can be leveraged to balance multimodal learning by emphasizing the less utilized modality with a larger weight. Setting $\omega_A, \omega_B = 0$ disables distillation, reverting the learning objective to Equation 2.

Overall, the MIND framework produces a smaller, optimized version of the original multimodal model. This compressed model can make predictions using both multimodal (when both modalities are present) and

unimodal (single modality) inputs. By introducing weighted unimodal ensemble knowledge distillation, the MIND framework enhances and balances modality representation learning, significantly improving predictive performance for both multimodal and unimodal inputs. We provide a pseudo-code implementation of the MIND framework for training an enhanced, compact multimodal student network based on the MIND framework for two modalities in Algorithm 1.

---

**Algorithm 1** MIND Framework

---

**Require:** Multimodal training dataset $\mathcal{D}_{AB_{train}}$, Multimodal validation dataset $\mathcal{D}_{AB_{val}}$, Modality A training dataset $\mathcal{D}_{A_{train}}$, Modality A validation dataset $\mathcal{D}_{A_{val}}$, Modality B training dataset $\mathcal{D}_{B_{train}}$, Modality A training validation $\mathcal{D}_{B_{train}}$, Multimodal Model $M_{AB}$, Loss function $\mathcal{L}_{\text{MIND}}$, Number of epochs $N$, Optimizer $O$, Weighting coefficients $w_A$, $w_B$

**Ensure:** Trained multimodal model $M_{AB}$

    /* Train $T$ unimodal teacher models per modality */

    /* Create ensemble of unimodal teachers per modality */

1: $E_A = [T_0^A,...,T_T^A]$

2: $E_B = [T_0^B,...,T_T^B]$

    /* Train multimodal student model $M_{AB}$ */

3: Initialize model parameters and add classification heads to modality encoders ($M_A$, $M_B$)

4: **for** epoch = 1 to $N$ **do**

5:     Shuffle the training data $\mathcal{D}_{AB_{train}}$

6:     **for** minibatch $b$ in $\mathcal{D}_{AB_{train}}$ **do**

7:         Forward pass: Compute predictions with all classification heads $\hat{y}_{AB}$, $\hat{y}_A$, $\hat{y}_B$, and teacher ensembles $\hat{y}_{E_A}$, $\hat{y}_{E_B}$ for minibatch $b$ using models $M$, $M_A$, $M_B$, $E_A$ and $E_B$

8:         Compute loss components $\mathcal{L}_{S_{AB}}(\hat{y}_{AB}, y)$, $\mathcal{L}_{S_A}(\hat{y}_A, y)$, $\mathcal{L}_{S_B}(\hat{y}_B, y)$, $\mathcal{L}_{EKD_A^U}(\hat{y}_A, \hat{y}_{E_A})$ and $\mathcal{L}_{EKD_B^U}(\hat{y}_B, \hat{y}_{E_B})$ for minibatch $b$ following Equation 2 and Equation 4

9:         Compute $\mathcal{L}_{\text{MIND}}(\mathcal{L}_{S_{AB}}, \mathcal{L}_{S_A}, \mathcal{L}_{S_B}, \mathcal{L}_{EKD_A^U}, \mathcal{L}_{EKD_B^U}, w_A, w_B)$ for minibatch $b$ following Equation 6

10:         Backward pass: Compute gradients of the loss with respect to model $M_{AB}$ parameters

11:         Update model parameters using optimizer $O$

12:     **end for**

13: **end for**=0

---

## 4 Experiments

**Datasets.** To evaluate our approach, we focus on two clinical prediction tasks: predicting clinical conditions ($L = 25$, Equation 3), and predicting in-hospital mortality after a 48-hour ICU stay ($L = 1$, Equation 3), using Chest X-Ray (CXR) images and clinical time-series data extracted from the patient's Electronic Health Records (EHR). We use chest X-ray images from MIMIC-CXR (Johnson et al., 2019b), where each image ($\mathbf{x}_A$) is replicated across three channels, yielding $\mathbf{x}_A \in \mathbb{R}^{224 \times 224 \times 3}$. Associated clinical time-series data ($\mathbf{x}_B$) is obtained from MIMIC-IV (Johnson et al., 2023), with dimensions $\mathbf{x}_B \in \mathbb{R}^{t \times 76}$, where $t$ represents the number of time-steps based on the patient's ICU stay duration, and 76 denotes the number of pre-processed features per time-step. We follow the dataset splits and multimodal architecture from the work of Hayat et al. (2022), where $f_A$ is parameterized by a ResNet-34, $f_B$ is parameterized as a two-layer LSTM and the fusion model $g_{AB}$ is parameterized as an LSTM layer, referred to as MedFuse. We use the multimodal dataset for training the MIND framework and baselines, containing samples with both modalities present. For the clinical conditions task, the dataset comprises 7,728 training, 877 validation, and 2,161 test samples. For in-hospital mortality prediction, it consists of 4,885 training, 540 validation, and 1,373 test samples. When training unimodal models, we utilize all available data for each modality. Specifically, the CXR dataset comprises 124,671 training, 15,282 validation, and 36,625 test samples, while the EHR dataset contains 42,628 training, 4,802 validation, and 11,914 test samples.

**Ensembles.** In the MIND framework, we first pre-train the unimodal teachers using the available unimodal datasets. To ensure diversity, we train multiple models for each modality, following the architectures outlined in previous work (Hayat et al., 2022). For chest X-ray images, we train ResNet-34, ResNet-18, and ResNet-10

models. Similarly, for clinical time-series data, we train 2-layer, 3-layer, and 4-layer LSTM networks. The top three performing models from each modality are then combined to create ensembles. These ensembles of unimodal teachers are utilized to distill knowledge, as described in Equation 6, into a smaller randomly initialized multimodal student model. This student model consists of a ResNet-10 and a 2-layer LSTM, serving as encoders for chest X-ray and time-series data, respectively.

**Baselines.** In this work, we focus on joint fusion, such that both encoders are trained from scratch (randomly initialized), although previous work primarily focuses on fine-tuning pre-trained unimodal encoders (Huang et al., 2020). We believe that learning with randomly initialized encoders is more appropriate in our proposed setup for several reasons. First, randomly initialized encoders establish a consistent baseline for fair comparisons across models, ensuring that any performance improvements reflect the architecture and training methods rather than pre-trained weights. Additionally, starting with random initialization helps isolate the specific learning dynamics of our setup, avoiding the potential of skewing the performance as a result of using fine-tuned pre-trained encoders. Our approach also introduces unique elements that may not align with assumptions of previous work, such as avoiding the need for pre-training the student model, so we can assess how well the model learns from scratch to understand its full capabilities. Finally, using randomly initialized encoders eliminates biases from original training data associated with fine-tuning, allowing for a more objective evaluation of our method.

We compare the performance of MIND with the original multimodal model, MedFuse. The MIND model is three times smaller (in terms of learnable parameters) than MedFuse, as shown in Table A1. Further, we modify the original MedFuse architecture to adopt the loss introduced in Equation 2, denoted as MedFuse-3H. We also compare MIND to other related KD baselines, including TS (Wang et al., 2020a), MKE (Xue et al., 2021), and UME (Du et al., 2023). For MKE, we evaluate two models: MKE-CXR and MKE-EHR. Further details about the baseline models can be found in Appendix A.4. All MIND and baseline models trained for each specific task are all trained using the same dataset sizes, as described in Appendix A.2 and Appendix C.1.

**Hyperparameter Tuning.** All models, including baselines, are trained for a maximum of 300 epochs with early stopping after 40 epochs. We use a batch size of 16 and the Adam optimizer across all experiments. Hyperparameter tuning involves optimizing the learning rate and weighting parameters for MIND and all baselines. Further details on hyperparameter tuning are provided in Appendix A. We select the best models based on the checkpoint yielding the highest Area Under the Receiving Operating Characteristic (AUROC) on the validation set and present the results on the test set in terms of AUROC and Area Under the Precision-Recall Curve (AUPRC). We also report 95% confidence intervals with 1,000 iterations using the bootstrapping method (Efron & Tibshirani, 1994). For reproducibility, our code is included in Appendix A.5 as we aim to make it publicly available, following the pseudo-code implementation of the MIND framework in Algorithm 1.

We conduct our experiments on a shared high-performance computing cluster equipped with Nvidia A100 GPUs. For the clinical conditions task, training the models takes less than 20 hours (120-150 epochs), while for in-hospital mortality prediction, the models are trained in under two hours.

## 5 Results

### 5.1 Multimodal fusion performance results

Table 1 compares the performance of multimodal baselines with our proposed model on the test set. The MIND framework achieves the highest AUROC and AUPRC across both tasks, outperforming all baselines by over 1.4% AUROC and 2.3% AUPRC for clinical conditions prediction, and 1.6% AUROC and 1.4% AUPRC for in-hospital mortality prediction. Notably, MIND surpasses MedFuse and MedFuse-3H by over 3.4% AUROC and 5.4% AUPRC for clinical conditions, and by over 2.4% AUROC and 3.7% AUPRC for in-hospital mortality. Validation set results are in Appendix B.1. The results indicate that the MIND framework excels at compressing knowledge from unimodal teacher models while enabling unimodal predictions for unimodal samples — an advantage not present in the compared frameworks.

Table 1: **Multimodal fusion performance results.** Performance results on the multimodal test set for the MIND model and all baseline models (MedFuse, MedFuse-3H, TS, MKE-CXR, MKE-EHR, and UME). The best results are highlighted in bold.

| Model | Clinical Conditions | | In-hospital Mortality | |
|---|---|---|---|---|
| | AUROC | AUPRC | AUROC | AUPRC |
| MedFuse (Hayat et al., 2022) | 0.748 (0.721, 0.774) | 0.447 (0.402, 0.495) | 0.816 (0.785, 0.847) | 0.468 (0.398, 0.546) |
| MedFuse-3H | 0.750 (0.724, 0.777) | 0.452 (0.410, 0.500) | 0.820 (0.787, 0.851) | 0.461 (0.392, 0.541) |
| MKE-CXR (Xue et al., 2021) | 0.711 (0.681, 0.739) | 0.412 (0.372, 0.458) | 0.707 (0.667, 0.747) | 0.308 (0.255, 0.377) |
| MKE-EHR (Xue et al., 2021) | 0.744 (0.717, 0.770) | 0.447 (0.405, 0.495) | 0.827 (0.795, 0.856) | 0.491 (0.422, 0.567) |
| UME (Du et al., 2023) | 0.767 (0.741, 0.793) | 0.489 (0.450, 0.544) | 0.818 (0.787, 0.849) | 0.491 (0.419, 0.569) |
| TS (Wang et al., 2020a) | 0.768 (0.742, 0.794) | 0.483 (0.439, 0.532) | 0.828 (0.797, 0.857) | 0.483 (0.414, 0.555) |
| MIND (Ours) | **0.782** (0.757, 0.807) | **0.506** (0.460, 0.556) | **0.844** (0.815, 0.872) | **0.505** (0.433, 0.587) |

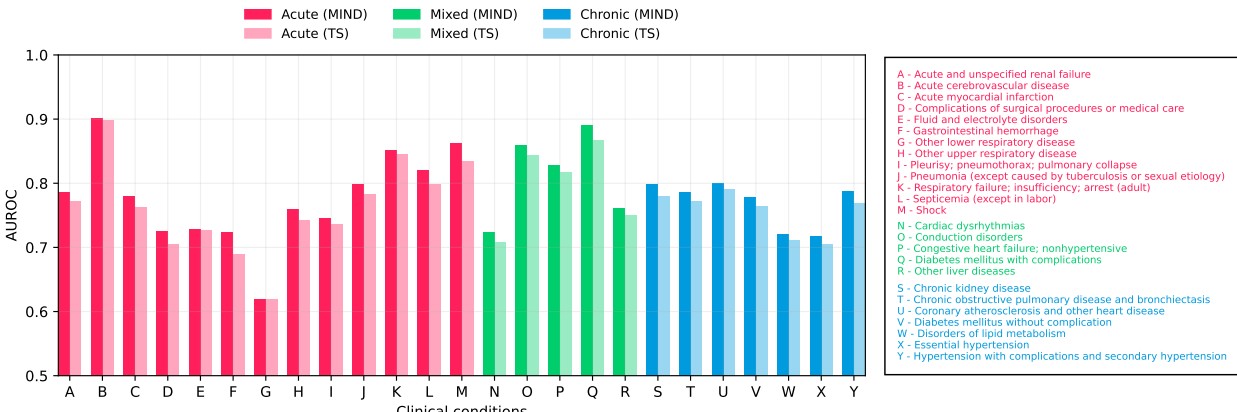

Figure 2: **Label-wise AUROC Performance**. Comparison of AUROC performance between the MIND model and the best baseline (TS) for each clinical condition label and group (acute, mixed, and chronic).

Figure 2 shows the label-wise AUROC performance of the MIND model with TS, the best baseline. MIND outperforms TS in overall multimodal AUROC and AUPRC metrics and shows superior AUROC across all individual and group labels: 0.777 vs. 0.763 for acute conditions, 0.813 vs. 0.797 for mixed conditions, and 0.770 vs. 0.756 for chronic conditions. Similarly, for AUPRC (Figure B1, Appendix B.2), MIND scores 0.448 vs. 0.422 for acute conditions, 0.592 vs. 0.567 for mixed, and 0.551 vs. 0.534 for chronic conditions.

## 5.2 Quality of unimodal representations

We also evaluate the quality of the unimodal representations. The test set results are summarized in Table 2 and Table 3 for the clinical conditions and in-hospital mortality tasks, respectively. Additional results on the validation set are provided in Appendix B.3. For clinical conditions prediction using X-ray images, the MIND model achieves a superior AUROC (0.709 vs. 0.663) and AUPRC (0.409 vs. 0.349) compared to MedFuse-3H. Similarly, for time-series data, the MIND model outperforms MedFuse-3H with an AUROC of 0.746 vs. 0.715 and an AUPRC of 0.451 vs. 0.404. In the in-hospital mortality task, the MIND model improves chest X-ray model performance by over 10% compared to MedFuse-3H while maintaining similar performance for clinical time series. These results demonstrate that the MIND model not only enhances multimodal performance but also ensures unimodal performance is on par with unimodal models trained on much larger datasets (124,671 for CXR and 42,628 for EHR vs. 7,728 paired CXR-EHR samples).

Table 2: **Unimodal performance results for the clinical conditions task.** Evaluation of unimodal and multimodal models per modality on the multimodal test set, including the number of training samples per model. Note that unimodal models are trained on significantly larger datasets compared to the fusion models, which are trained on the multimodal set. The best results are highlighted in bold.

| Model | Training set | Chest X-Ray images | | Clinical time series | |
|---|---|---|---|---|---|
| | | AUROC | AUPRC | AUROC | AUPRC |
| **Multimodal** | | | | | |
| MedFuse-3H | 7,728 | 0.663 (0.633, 0.693) | 0.349 (0.313, 0.391) | 0.715 (0.687, 0.743) | 0.404 (0.365, 0.449) |
| MIND (Ours) | 7,728 | 0.709 (0.680, 0.738) | 0.409 (0.370, 0.455) | **0.746** (0.719, 0.772) | **0.451** (0.408, 0.499) |
| **Unimodal** | | | | | |
| ResNet-34 | 124,671 | 0.704 (0.674, 0.733) | 0.400 (0.361, 0.445) | - | - |
| ResNet-10 | 124,671 | **0.710** (0.681, 0.740) | **0.413** (0.373, 0.459) | - | - |
| 2-layer LSTM | 42,628 | - | - | 0.744 (0.717, 0.771) | 0.448 (0.406, 0.496) |
| 4-layer LSTM | 42,628 | - | - | 0.742 (0.715, 0.768) | 0.447 (0.404, 0.495) |

Table 3: **Unimodal performance results for the in-hospital mortality prediction task.** Evaluation of unimodal and multimodal models per modality on the multimodal test set, including the number of training samples per model. Note that unimodal models are trained on significantly larger datasets compared to the fusion models, which are trained on the multimodal set. The best results are highlighted in bold.

| Model | Training set | Chest X-ray images | | Clinical time series | |
|---|---|---|---|---|---|
| | | AUROC | AUPRC | AUROC | AUPRC |
| **Multimodal** | | | | | |
| MedFuse-3H | 4,885 | 0.572 (0.529, 0.617) | 0.201 (0.164, 0.251) | 0.827 (0.794, 0.857) | 0.480 (0.406, 0.555) |
| MIND (Ours) | 4,885 | **0.690** (0.648, 0.732) | **0.290** (0.236, 0.358) | **0.830** (0.795, 0.859) | **0.502** (0.427, 0.577) |
| **Unimodal** | | | | | |
| ResNet-34 | 124,671 | 0.681 (0.638, 0.724) | 0.276 (0.226, 0.346) | - | - |
| ResNet-10 | 124,671 | 0.682 (0.643, 0.721) | 0.259 (0.214, 0.318) | - | - |
| 2-layer LSTM | 42,628 | - | - | **0.830** (0.801, 0.858) | 0.489 (0.425, 0.569) |
| 4-layer LSTM | 42,628 | - | - | 0.823 (0.790, 0.853) | 0.495 (0.418, 0.569) |

These results highlight the MIND framework's ability to significantly enhance both multimodal and unimodal performance. The unimodal encoders can be utilized with unimodal samples, achieving performance comparable to that of unimodal encoders trained on larger datasets. Unlike related work, our multimodal compression framework uniquely incorporates the use of unimodal encoders.

### 5.3 Ablation study I: sensitivity analysis

We conduct ablation studies to understand the impact of each component in Equation 6. To evaluate the benefits of ensembling, we experiment with both single-teacher KD and EKD, using three models per modality ensemble. Specifically, we consider six settings:

1. Supervised learning as in previous work (Hayat et al., 2022), without knowledge distillation.

2. Supervised learning with modified loss (Equation 2), without knowledge distillation.

3. Supervised learning with unweighted unimodal single-teacher KD. Specifically, we set $\omega_A, \omega_B = 1$ in Equation 6 and use a single unimodal model as the modality teacher.

4. Supervised learning with weighted unimodal single-teacher KD. We apply Equation 6 using a single unimodal model as the modality teacher.

Table 4: **Ablation study on MIND for multimodal and unimodal performance in the clinical conditions task**. Fusion and unimodal performance on the multimodal test set for MIND with different loss components. Each setting indicates the components used in the modified loss for training and the type of knowledge distillation performed. The best results are highlighted in bold.

| # | $\mathcal{L}_{S_{AB}}$ | $\mathcal{L}_{S_{A/B}}$ | $\mathcal{L}_{KD^U_{A/B}}$ | $\mathcal{L}_{EKD^U_{A/B}}$ | $\omega_{A/B}$ | Fusion | | Chest X-Ray | | Time Series | |
|---|---|---|---|---|---|---|---|---|---|---|---|
| | | | | | | AUROC | AUPRC | AUROC | AUPRC | AUROC | AUPRC |
| 1 | ✓ | | | | | 0.743 (0.716, 0.769) | 0.440 (0.398, 0.489) | - | - | - | - |
| 2 | ✓ | ✓ | | | | 0.748 (0.721, 0.774) | 0.449 (0.407, 0.498) | 0.674 (0.644, 0.705) | 0.364 (0.328, 0.407) | 0.713 (0.685, 0.740) | 0.405 (0.366, 0.450) |
| 3 | ✓ | ✓ | ✓ | | | 0.761 (0.735, 0.787) | 0.472 (0.428, 0.520) | 0.686 (0.657, 0.716) | 0.381 (0.344, 0.423) | 0.730 (0.702, 0.756) | 0.427 (0.386, 0.474) |
| 4 | ✓ | ✓ | | ✓ | | 0.766 (0.740, 0.791) | 0.476 (0.432, 0.525) | 0.689 (0.659, 0.718) | 0.383 (0.345, 0.427) | 0.734 (0.706, 0.761) | 0.428 (0.386, 0.474) |
| 5 | ✓ | ✓ | ✓ | | ✓ | 0.778 (0.753, 0.803) | 0.498 (0.453, 0.548) | 0.695 (0.665, 0.725) | 0.394 (0.356, 0.438) | 0.738 (0.711, 0.764) | 0.437 (0.396, 0.484) |
| 6 | ✓ | ✓ | | ✓ | ✓ | **0.782** (0.757, 0.807) | **0.506** (0.460, 0.556) | **0.709** (0.680, 0.738) | **0.409** (0.370, 0.455) | **0.746** (0.719, 0.772) | **0.451** (0.408, 0.499) |

5. Supervised learning with unweighted unimodal EKD. We set $\omega_A, \omega_B = 1$ in Equation 6 and use an ensemble of three teacher models.

6. MIND: supervised learning with weighted unimodal EKD (Equation 6).

Table 4 shows the MIND setting results for the clinical conditions task, while Table B4 presents results for the in-hospital mortality task, both in terms of AUROC and AUPRC on the test set. Validation set results for both tasks can be found in Appendix B.4. We observe that adding supervised loss terms for both modalities slightly improves AUROC (from 0.743 to 0.748) and AUPRC (0.440 to 0.449). Incorporating the unimodal unweighted distillation component further improves AUROC and AUPRC to 0.761 and 0.472, respectively, with ensembling boosting these metrics to 0.766 and 0.476. Adding weighting coefficients to the distillation components significantly enhances performance on both multimodal and unimodal predictions. This demonstrates the benefits of distilling knowledge from pre-trained unimodal networks and highlights the importance of the weighting parameters in Equation 6. Collectively, all these factors collectively enhance the multimodal model, enabling unimodal predictions and achieving superior overall performance.

## 5.4 Ablation study II: balancing multimodal learning with $w_A$ and $w_B$

We conduct a hyperparameter sensitivity analysis for the weighting parameters $\omega_A$ and $\omega_B$ introduced in Equation 6. These parameters help the model focus on unimodal representation learning, thereby improving the quality of unimodal predictions, as shown in Section 5.2. We evaluate their impact on enhancing the balance of multimodal learning using the conditional utilization rate per modality (Wu et al., 2022b), which we adopt to characterize modality usage in the multimodal deep neural network. Specifically, we modify **u** for joint fusion multimodal neural networks as follows:

$$\mathbf{u_A} = \frac{A(\hat{\mathbf{y}}_{AB}) - A(\hat{\mathbf{y}}_B)}{A(\hat{\mathbf{y}}_{AB})}, \mathbf{u_B} = \frac{A(\hat{\mathbf{y}}_{AB}) - A(\hat{\mathbf{y}}_A)}{A(\hat{\mathbf{y}}_{AB})},$$

where $A$ and $B$ are two modalities without loss of generality, $A(\cdot)$ denotes a classification accuracy metric, **$u_A$** computes the conditional utilization rate for modality A, and **$u_B$** for modality B. The conditional utilization rate measures the marginal contribution of each modality to the fusion model. Following Wu et al. (2022b), $d_{util}$ is defined as the difference between conditional utilization rates: $d_{util} = \mathbf{u_A} - \mathbf{u_B}$. This allows for the assessment of imbalanced modality usage within the multimodal fusion model. Specifically, $d_{util} \in \mathbb{R}\ s.t. -1 \leq d_{util} \leq 1$, with extreme values indicating imbalanced modality usage.

We use the validation AUROC as the accuracy metric and test six settings: (i) $\omega_{cxr} = \omega_{ehr} = 0$, (ii) $\omega_{cxr} \ll \omega_{ehr}$, (iii) $\omega_{cxr} < \omega_{ehr}$, (iv) $\omega_{cxr} = \omega_{ehr}$, (v) $\omega_{cxr} > \omega_{ehr}$, and (vi) $\omega_{cxr} \gg \omega_{ehr}$. Figure 3 shows the conditional utilization rates and their differences for the clinical conditions task. Table B7 details the specific values of the hyper-parameters. In the setting where $\omega_{cxr} = \omega_{ehr} = 0$, the fusion model tends to overfit the EHR modality, with utilization rate differences reaching up to 10%. Assigning significantly different weights to the modalities, as seen in the $\omega_{cxr} \gg \omega_{ehr}$ and $\omega_{cxr} \ll \omega_{ehr}$ settings, increases the conditional utilization rate of the modality with the larger weight while decreases it for the other, thereby affecting the balance of utilization rates during training. This effect can be leveraged to achieve more balanced multimodal learning,

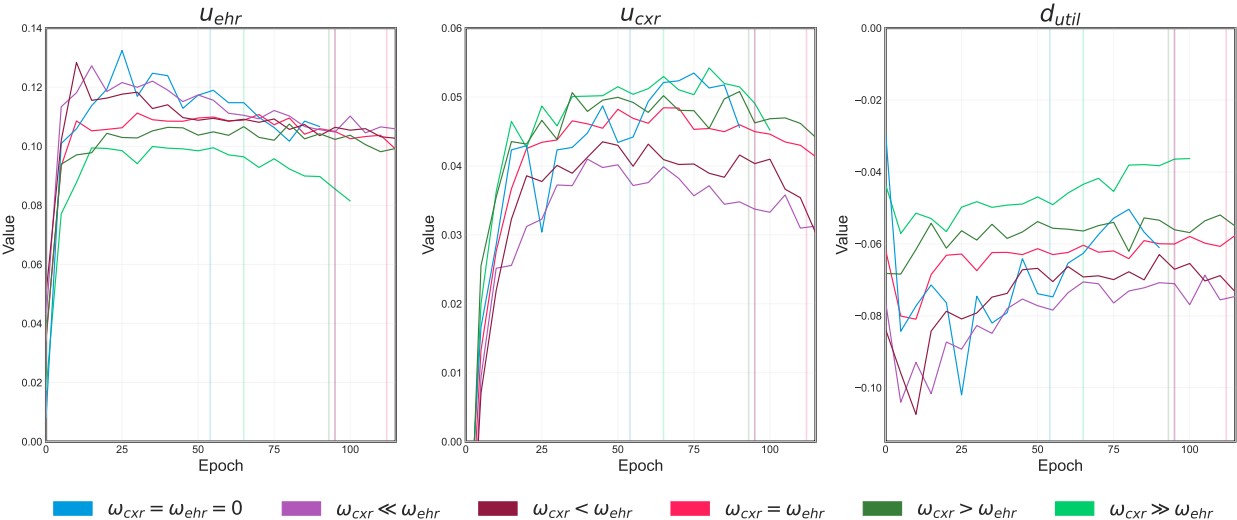

Figure 3: **Characterization of modality utilization in multimodal learning for the clinical conditions task.** The line graphs depict the conditional utilization rates ($u_{cxr}, u_{ehr}$) and their difference ($d_{util} = u_{cxr} - u_{ehr}$) every 5 training epochs for the six settings. Vertical lines indicate the epoch with the best AUROC performance for each model on the validation set.

as seen in the $\omega_{cxr} \gg \omega_{ehr}$ setting, which emphasizes the under-utilized CXR modality. Here, the EHR modality utilization never exceeds 10%, and the difference between the conditional utilization rates remains around 5%, the smallest observed. A smaller absolute value of $d_{util}$ indicates more balanced multimodal learning. Additionally, in this setting, the utilization is consistent with a smoother $u_{cxr}$ line. This indicates that assigning a larger weight to the under-utilized modality results in more balanced multimodal learning, reducing overfitting of the dominant modality and increasing the use of the weaker modality. However, it's important to note that more balanced multimodal training does not always equate to better fusion performance; rather, it is related to improved unimodal encoder performance.

## 5.5 Additional Results on Multimodal Benchmark Datasets

The results in Section 5.1 demonstrate the utility of the MIND framework in clinical settings, which is the primary motivation for our study, particularly for multilabel and binary clinical tasks, using a state-of-the-art architecture as the base model. To evaluate the generalization and applicability of the MIND framework across diverse multimodal datasets, tasks, and architectures, we conducted additional validation on three multimodal, multiclass benchmark datasets: CREMA-D (Cao et al., 2014), S-MNIST (Khacef et al., 2019), and LUMA (Bezirganyan et al., 2024). We report the best accuracy results for each model in Table 5. For these experiments, the feature encoders consist of ResNet-3/6 models, with the output from each encoder concatenated and used as input for the fusion model, which is a linear layer. Additional implementation details, such as dataset descriptions, the multimodal architectures employed, and their sizes, can be found in Appendix C.

As shown in Table 5, the MIND model consistently outperforms all multimodal baselines across all datasets. Additionally, the performance of the unimodal encoders either surpasses or is comparable to that of the baselines. Note that UME does not involve any multimodal training; it simply averages the predictions of the unimodal models to compute the multimodal prediction. We observe that the MIND model improves multimodal prediction by an average of 1.5% and boosts the performance of the weaker modality by approximately 5.2% across tasks.

Table 5: **Multimodal fusion performance results on multiclass classification tasks.** Performance results on the multimodal test set for the MIND model and all baselines (Vanilla, Vanilla-3H, TS, MKE-CXR, MKE-EHR, and UME) across the multiclass classification tasks defined by the CREMA-D, S-MNIST, and LUMA datasets. The best results are highlighted in bold.

| Model | CREMA-D | | | S-MNIST | | | LUMA | | |
|---|---|---|---|---|---|---|---|---|---|
| | Audio-Video | Audio | Video | Audio-Image | Image | Audio | Audio-Image | Audio | Image |
| Vanilla | 0.599 | 0.482 | 0.273 | 0.970 | 0.967 | 0.417 | 0.847 | 0.385 | 0.280 |
| Vanilla-3H | 0.616 | 0.545 | 0.315 | 0.966 | 0.981 | 0.604 | 0.870 | 0.539 | 0.599 |
| MKE-AUDIO (Xue et al., 2021) | 0.612 | 0.555 | 0.235 | 0.664 | 0.515 | 0.521 | 0.715 | 0.456 | 0.346 |
| MKE-VISUAL (Xue et al., 2021) | 0.601 | 0.442 | 0.382 | 0.966 | 0.965 | 0.379 | 0.815 | 0.429 | 0.428 |
| UME (Du et al., 2023) | 0.620 | **0.561** | 0.450 | 0.898 | **0.998** | 0.699 | 0.903 | **0.759** | 0.606 |
| TS (Wang et al., 2020a) | 0.625 | 0.528 | 0.274 | 0.970 | 0.930 | 0.473 | 0.894 | 0.610 | 0.509 |
| MIND (Ours) | **0.641** | 0.552 | **0.461** | **0.982** | 0.981 | **0.772** | **0.921** | 0.721 | **0.677** |

Overall, the results in Table 1 and Table 5 demonstrate that the MIND model significantly outperforms all baseline models across datasets, tasks, and architectures. These findings highlight the advantages of MIND over other multimodal training methods in terms of both multimodal and unimodal encoder performance for binary, multiclass, and multilabel tasks. Specifically, we have validated our proposed framework on four multimodal datasets (MIMIC-IV-CXR, CREMA-D, S-MNIST and LUMA), five tasks, four architectures, and two fusion methods. MIND consistently outperforms all multimodal training baselines.

## 6 Discussion

Recent multimodal learning research highlights the challenges of learning from multiple modalities (Wu et al., 2022b; Wang et al., 2020b). While increasing model size is sometimes feasible, it is not always desirable. Additionally, multimodal models often overfit to one modality, under-utilizing others. In this work, we propose MIND, a simple yet effective KD framework that distills knowledge from pre-trained unimodal teachers to a smaller multimodal student, enhancing both multimodal and unimodal predictive power. Our experimental results demonstrate significant benefits from this approach, likely due to unimodal encoders leveraging larger training datasets and learning better representations than those trained on the comparably smaller multimodal clinical datasets. To further improve ensemble impact, we hypothesize that stronger teachers and potentially a larger number of teachers are needed (Hinton et al., 2015). Notably, our framework is architecture-agnostic and can be easily adapted to similar settings in other application domains.

While Buciluă et al. (2006), Fukuda et al. (2017), Asif et al. (2020), Li et al. (2021), and Wu et al. (2022a) combine ensemble learning and offline response-based knowledge distillation to train compact networks, their approaches primarily focus on the unimodal networks. In contrast, we are concerned with the particularities of multimodal joint fusion networks, both in performance and size, thus we propose incorporating and weighting modality-specific ensembles during student knowledge distillation as a distinctive feature. Despite methodological differences, MIND similarly avoids the need for a large ensemble of classifiers, which demands significant resources during inference. In addition, the MIND framework can be used to address imbalanced multimodal learning during training and leverages modality encoders to handle unimodal samples, achieving predictive performance for unimodal samples that is on par with powerful unimodal models.

The introduction of the new learning objective (Equation 6) in our proposed approach does not compromise its applicability to real-world scenarios, particularly regarding its scalability as the number of modalities, $M$, increases. For two modalities, as defined in Equation 6, the loss consists of five components: three supervised learning losses and two weighted unimodal ensemble knowledge distillation losses. For $M = 3$, the proposed loss would contain seven components: four supervised losses and three weighted unimodal ensemble knowledge distillation losses, incorporating one additional supervised loss and one knowledge distillation loss for the new modality. Generalizing to $M$ modalities, the proposed loss would have $2M + 1$ loss terms, i.e., $\mathcal{O}(M)$, making our approach linearly scalable for practical settings.

We customize the framework for multilabel classification, addressing a less explored area in knowledge distillation, which typically focuses on multiclass classification. Multilabel classification is common in clinical prediction tasks, where multiple labels may be present in a single sample. We also evaluate our framework on a binary classification task. We demonstrate its effectiveness in two relevant clinical tasks: multilabel clinical conditions prediction and binary in-hospital mortality prediction, using the publicly available MIMIC-CXR and MIMIC-EHR datasets. In addition, we assess its performance on multimodal multiclass benchmark datasets, providing a comprehensive validation of our proposed framework across multilabel, multiclass, and binary classification tasks. Our code is publicly available to ensure reproducibility and facilitate future comparisons (see Appendix A.5).

**Limitations.** Since our evaluation is limited to two clinical tasks, further experiments with additional clinical tasks and datasets are necessary. However, the scarcity of real-world multimodal medical datasets that include both images and clinical time-series data (heterogeneous modalities) poses a significant challenge. Additionally, while our solution can help balance multimodal learning, it cannot ensure complete balance if one modality entirely dominates. In such cases, any multimodal approach may be inefficient. In future work, we aim to explore larger ensembles and employ stronger teachers. Despite the advantages of a compressed multimodal network during deployment, significant computational resources are still required during training. Specifically, our framework is embedded within the offline KD paradigm. The current state of the art in this KD paradigm establishes a two-step method: first, the training of various teacher models, followed by a second step for training the student model. This two-step process is essential to offline KD, which requires significant computational resources. However, to address this increase in resource demand, a distinctive feature of MIND is our proposal to ensemble the teacher models instead of discarding sub-optimal ones, thereby reducing computational waste. Finally, considering the societal impact, while multimodal networks may enhance overall performance in the test cohort, further research is needed to assess their fairness and generalizability across patient subcohorts and at the instance level.

# 7 Broader Impact Statement

**Potential Risks of Malicious Usage of Medical Models.** The deployment of machine learning models in healthcare carries significant risks if misused. Malicious actors could potentially exploit these models to manipulate medical diagnoses, treatment plans, or patient data. For instance, altering a model's predictions could lead to incorrect diagnoses or inappropriate treatments, posing severe health risks to patients. In addition, the unauthorized access and manipulation of sensitive clinical data could lead to privacy breaches and identity theft.

**Implicit Biases Learned from Knowledge Distillation.** Machine learning models trained on clinical datasets are susceptible to inheriting and amplifying existing biases present in the training data. Knowledge distillation, a model training process where a smaller model learns from larger, pre-trained models, can propagate these biases. This can result in unequal treatment outcomes for different demographic groups, exacerbating health disparities. For instance, if the training data contains biases against certain ethnicities or genders, the distilled model may continue to exhibit these biases, leading to unfair treatment recommendations.

**Privacy Concerns with MIMIC-IV and MIMIC-CXR data.** The clinical datasets used in this research are MIMIC-IV (Johnson et al., 2023) and MIMIC-CXR (Johnson et al., 2019b). The authors would like to note that there are no privacy concerns related to these datasets. In particular, the MIMIC-IV and MIMIC-CXR datasets are de-identified and adhere to strict privacy regulations, ensuring that patient confidentiality is maintained (Johnson et al., 2024; 2019a). This allows researchers to utilize real-world clinical data for developing and testing machine learning models without compromising patient privacy.

While machine learning holds great promise for advancing healthcare delivery, it is critical to consider and address these broader impact concerns. Ensuring robust security measures, mitigating biases, and maintaining patient privacy are essential steps towards the responsible and ethical use of machine learning in healthcare.

# 8 Conclusion

Multimodal learning offers potential enhancements for clinical prediction tasks by leveraging cross-modality interactions. However, integrating multiple modalities can lead to increased network complexity and size, posing challenges for resource-constrained applications. Additionally, multimodal clinical data present hurdles such as modality heterogeneity and missing data. Overall, our work addresses these challenges through weighted ensemble knowledge distillation, providing a promising approach to enhance fusion networks for real-world multimodal clinical data. Finally, we demonstrate the generalizability of our proposed approach by validating it across three multimodal benchmark datasets, as well as two clinical tasks, two fusion methods, and various multimodal network architectures.

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

# A    Implementation details

## A.1    Model size

Table A1 shows the sizes of the different model architectures. MedFuse (Hayat et al., 2022) serves as the base architecture for our work and is provided as a reference. While it was originally proposed using ResNet-34 and 2-layer LSTM encoders, we adopt the same configuration for MIND (ResNet-10 and 2-layer LSTM) and all baselines for the sake of comparison. Notably, the MIND model is three times smaller than the original architecture while providing unimodal classification capabilities.

Table A1: **Size of multimodal models.** We provide a summary of the number of trainable parameters for each multimodal architecture. We use ResNet-10 and 2-layer LSTM encoders for the student network within the proposed MIND framework and for all baselines.

| Model | No. of trainable parameters |
|---|---|
| ResNet-34 and 2-layer LSTM (Hayat et al., 2022) | 23.9 M |
| ResNet-10 and 2-layer LSTM (MIND) | 7.5 M |

## A.2    Clinical dataset

In our experimental setup, we use two publicly available datasets for two clinical prediction tasks: (i) multilabel clinical conditions prediction and (ii) in-hospital mortality prediction. For modality A, we use chest X-ray images (CXR) extracted from the MIMIC-CXR dataset (Johnson et al., 2019b). For modality B, we use electronic health record (EHR) time-series data extracted from the MIMIC-IV dataset (Johnson et al., 2023). We link the modalities following Hayat et al. (2022). The dataset comprises all clinical time-series that have an associated chest X-ray image, ensuring that both modalities are present for each sample. We summarize the dataset in Table A2.

Table A2: **Summary of dataset.** Size, shape and composition of the datasets used to train, validate, and evaluate the unimodal and multimodal models. The chest X-ray images, where a single image $\mathbf{x}_A \in \mathbb{R}^{224 \times 224 \times 3}$, are represented by dataset $\mathcal{D}_A$, and the clinical time-series data, such that a sample $\mathbf{x}_B \in \mathbb{R}^{t \times 76}$ where $t$ is the number of time-steps based on patient's stay in the intensive care unit and 76 is the number of pre-processed features per time-step, are represented by $\mathcal{D}_B$. For the multimodal datasets, we paired samples from $\mathcal{D}_A$ and $\mathcal{D}_B$, using the resulting datasets for our experiments.

| Dataset | Training | Validation | Test | Shape, composition |
|---|---|---|---|---|
| **Unimodal** | | | | |
| CXR ($\mathcal{D}_A$) | 124671 | 8813 | 20747 | $\mathbf{x}_A \in \mathbb{R}^{224 \times 224 \times 3}$ |
| EHR ($\mathcal{D}_B$) | 42628 | 4802 | 11914 | $\mathbf{x}_B \in \mathbb{R}^{t \times 76}$ |
| **Multimodal** | | | | |
| Clinical conditions | 7756 | 882 | 2166 | $\mathcal{D}_A \cap \mathcal{D}_B$ |
| In-hospital mortality | 4885 | 540 | 1373 | $\mathcal{D}_A \cap \mathcal{D}_B$ |

## A.3    MIND architecture & implementation

We use MedFuse (Hayat et al., 2022) as the multimodal baseline for our experimental setup. The original implementation of MedFuse employs an LSTM-based fusion module that processes a sequence of modality representations provided by a ResNet-34 encoder for the chest X-ray images and a 2-layer LSTM encoder for the clinical time-series data. In the MIND framework, we reduce the size of the image encoder to a ResNet-10 model to achieve model compression. In our experiments, we adhere to the hyperparameters used in the original MedFuse implementation for the randomly initialized multimodal fusion network.

For the MIND model and all baselines, we perform random hyper-parameter tuning of the learning rates and weighting coefficients where applicable. We conduct a minimum of 50 runs per model on each task. For the unimodal baselines, we randomly select ten different learning rates and use the best-performing model based on the epoch with the highest AUROC on the validation set. All hyper-parameters are summarized in Table A3.

Table A3: **Model hyper-parameters.** Description of the hyper-parameters used in our experimental setup. We follow those suggested by Hayat et al. (2022) for the multimodal model with randomly initialized encoders and baselines. For the unimodal encoders, we evaluate different learning rates and select the best one based on the epoch with the highest AUROC on the validation set.

| Hyper-parameter | Value/s |
| --- | --- |
| Batch size | 16 |
| Drop out | 0.3 |
| Epochs | 300 |
| Early stopping | Patience = 40 |
| Learning rate - multimodal/baselines | $[1 \times 10^{-3}, ..., 1 \times 10^{-5}]$ |
| Learning rate - unimodal CXR | $[1 \times 10^{-4}, ..., 1 \times 10^{-6}]$ |
| Learning rate - unimodal EHR | $[1 \times 10^{-4}, ..., 1 \times 10^{-6}]$ |

## A.4 Multimodal Baselines

In alignment with our proposed response-based knowledge distillation framework, MIND, we adapt the baselines for multilabel classification according to their corresponding papers, including MedFuse (Hayat et al., 2022), MedFuse-3H, MKE (Xue et al., 2021), UME (Du et al., 2023) and TS (Wang et al., 2020a). In our MIND framework and for all baselines, we save the best model checkpoint based on AUROC on the validation set during training and load the saved model to evaluate performance on the test set. Table A4 provides an overview of the loss functions proposed by each baseline for multimodal training.

**MedFuse** (Hayat et al., 2022) serves as the base architecture for all models and experiments in our work. It is a typical multimodal architecture proposed for medical tasks, featuring two modality encoders: a ResNet-34 for CXR images and a 2-layer LSTM for time-series data. The outputs from the encoders are concatenated and passed through an LSTM layer before the final multimodal classification head. To incorporate their methodology, we utilize the open-source code available at https://github.com/nyuad-cai/MedFuse. In our experiments, we perform learning rate tuning while adopting the default settings for the other hyper-parameters.

**MedFuse-3H**. Like most multimodal networks, the original MedFuse architecture does not include a classification head on the modality encoders, resulting in an exclusively multimodal output. In our MIND framework, the first step is to modify the multimodal network architecture to leverage the encoders for samples with absent modalities, thus avoiding the need of masking or imputation. We extend the original MedFuse architecture to include three classification heads (3H), naming it MedFuse-3H, and use it as a baseline for our approach. In our experiments, We perform learning rate tuning while adopting the default MedFuse settings for the other hyper-parameters.

**TS** (Wang et al., 2020a) proposes a response-based knowledge distillation framework where teacher models, trained with large datasets that include samples with missing modalities, transfer knowledge via soft labels to a multimodal student model. The multimodal student is trained using the soft labels from the unimodal teachers along with the supervision loss, as shown in Table A4. Originally proposed for multiclass classification, we adapt it for our multilabel classification framework. We perform random hyper-parameter tuning of the learning rate and the $\alpha, \beta$ distillation coefficients. Following Wang et al. (2020a), $\alpha, \beta \in \{0.0, 0.1, 0.2, 0.3, 0.4, 0.5, 0.6, 0.7, 0.8, 0.9\}$. We keep the other hyper-parameters consistent across models and baselines, using the default MedFuse settings.

**Multimodal Knowledge Expansion (MKE)** (Xue et al., 2021) proposes a response-based knowledge distillation framework composed of a unimodal teacher and a multimodal student. Originally designed to work with labeled and unlabeled data and evaluated on binary classification, multiclass classification, and segmentation tasks, we adapt it for our multilabel classification tasks framework. While it proposes the loss described in Table A4, we train the multimodal student network using a method equivalent to the equation in the table, following the authors' instructions. Since it proposes knowledge distillation (KD) from a single unimodal teacher to a multimodal student, we split this baseline into two: **MKE-EHR**, where we use the best EHR unimodal model as the teacher, and **MKE-CXR**, where we use the best CXR unimodal model as the teacher. We perform learning rate tuning in our experiments while adopting the default MedFuse settings for other hyper-parameters.

**Uni-modal Ensemble (UME)** (Du et al., 2023) proposes the aggregation of predictions from independently trained unimodal models for multimodal samples on a given task. The prediction for a multimodal instance is obtained by weighting the predictions of an ensemble of unimodal models (one model per modality). Following Du et al. (2023), we average the predictions from the unimodal models to provide the final prediction. Originally proposed for multiclass classification, we adapt it for our multilabel classification framework. No further training is required using this framework, apart from the independent training of the unimodal models.

Using the notation introduced earlier, we provide a summary of the loss functions proposed by the MIND framework and all baseline models.

Table A4: **Comparison of loss functions.** Using the notation introduced earlier, we provide a summary of the loss functions proposed by the MIND framework and all baseline models.

| Model | Loss function |
|---|---|
| MedFuse (Hayat et al., 2022) | $\mathcal{L}_{S_{AB}}$ |
| MedFuse-3H | $\mathcal{L}_{S_{AB}} + \mathcal{L}_{S_A} + \mathcal{L}_{S_B}$ |
| MKE-CXR (Xue et al., 2021) | $\mathcal{L}_{KD_A} + \gamma \times \mathcal{L}_{reg}$ |
| MKE-EHR (Xue et al., 2021) | $\mathcal{L}_{KD_B} + \gamma \times \mathcal{L}_{reg}$ |
| UME (Du et al., 2023) | $\mathcal{L}_{S_A}$ and $\mathcal{L}_{S_B}$ (independently trained unimodal models) |
| TS (Wang et al., 2020a) | $\mathcal{L}_{S_{AB}} + \alpha \times \mathcal{L}_{KD_{AB}^U} + \beta \times \mathcal{L}_{KD_{AB}^U}$ |
| MIND (ours) | $\mathcal{L}_{S_{AB}} + \mathcal{L}_{S_A} + \mathcal{L}_{S_B} + \omega_A \times \mathcal{L}_{EKD_A^U} + \omega_B \times \mathcal{L}_{EKD_B^U}$ |

### A.5 Code for reproducibility

To ensure the reproducibility of our results, we make our code available at: https://github.com/nyuad-cai/MIND

## B Additional results on clinical tasks

### B.1 Multimodal performance results on the validation set

Table B1 presents the performance results of the MIND model and all baseline models on the multimodal validation set for both clinical tasks.

### B.2 Label-wise AUPRC performance

Figure B1 compares the label-wise AUPRC performance of the MIND model with that of TS, the best baseline.

### B.3 Unimodal performance results on the validation set

Table B2 presents the unimodal performance results for the clinical conditions prediction task on the validation set. Table B3 provides the unimodal performance results for the in-hospital mortality prediction task on

Table B1: **Multimodal fusion performance results.** Performance results on the multimodal validation set for the MIND model and all baseline models (MedFuse, MedFuse-3H, TS, MKE-CXR, MKE-EHR, and UME). The best results are highlighted in bold.

| Model | Clinical Conditions | | In-hospital Mortality | |
|---|---|---|---|---|
| | **AUROC** | **AUPRC** | **AUROC** | **AUPRC** |
| MedFuse (Hayat et al., 2022) | 0.757 (0.716, 0.795) | 0.464 (0.397, 0.540) | 0.833 (0.778, 0.883) | 0.503 (0.395, 0.619) |
| MedFuse-3H | 0.760 (0.719, 0.798) | 0.460 (0.393, 0.536) | 0.835 (0.782, 0.884) | 0.438 (0.348, 0.566) |
| MKE-CXR (Xue et al., 2021) | 0.718 (0.673, 0.761) | 0.419 (0.360, 0.492) | 0.727 (0.664, 0.791) | 0.314 (0.239, 0.437) |
| MKE-EHR (Xue et al., 2021) | 0.753 (0.713, 0.791) | 0.456 (0.391, 0.531) | 0.850 (0.807, 0.899) | 0.544 (0.433, 0.672) |
| UME (Du et al., 2023) | 0.775 (0.735, 0.813) | 0.492 (0.425, 0.570) | 0.850 (0.797, 0.900) | 0.543 (0.423, 0.679) |
| TS (Wang et al., 2020a) | 0.778 (0.738, 0.815) | 0.490 (0.423, 0.568) | 0.846 (0.795, 0.890) | 0.563 (0.449, 0.667) |
| MIND (Ours) | **0.790** (0.752, 0.826) | **0.516** (0.446, 0.593) | **0.866** (0.817, 0.908) | **0.572** (0.455, 0.692) |

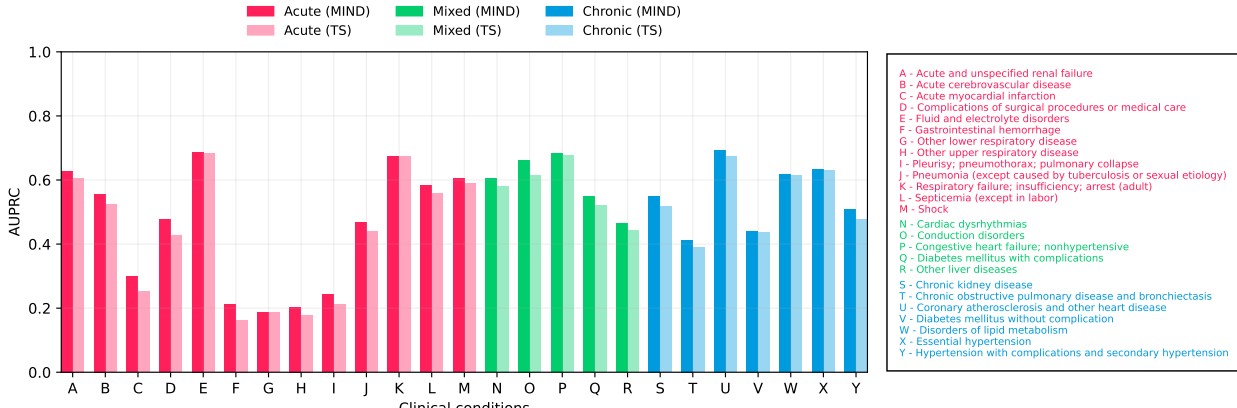

Figure B1: **Label-wise AUPRC Performance**. Comparison of AUPRC performance between the MIND model and the best baseline (TS) for each clinical condition label and group (acute, mixed, and chronic).

the validation set. Note that unimodal models are trained on much larger datasets compared to the fusion models, which are trained on the multimodal set.

Table B2: **Unimodal performance results for the clinical conditions task on the validation set.** Evaluation of unimodal and multimodal models per modality on the multimodal validation set, including the number of training samples per model.

| MODEL | TRAINING SET | CHEST X-RAY IMAGES | | CLINICAL TIME SERIES | |
|---|---|---|---|---|---|
| | | **AUROC** | **AUPRC** | **AUROC** | **AUPRC** |
| **Multimodal** | | | | | |
| MedFuse-3H | 7,728 | 0.679 (0.632, 0.724) | 0.371 (0.316, 0.439) | 0.713 (0.669, 0.754) | 0.401 (0.343, 0.472) |
| MIND (Ours) | 7,728 | 0.710 (0.665, 0.754) | 0.412 (0.353, 0.484) | 0.754 (0.714, 0.792) | 0.459 (0.394, 0.534) |
| **Unimodal** | | | | | |
| ResNet-34 | 124,671 | 0.714 (0.700, 0.727) | 0.442 (0.422, 0.464) | - | - |
| ResNet-10 | 124,671 | 0.719 (0.705, 0.732) | 0.450 (0.430, 0.472) | - | - |
| 2-layer LSTM | 42,628 | - | - | 0.762 (0.743, 0.780) | 0.420 (0.388, 0.456) |
| 4-layer LSTM | 42,628 | - | - | 0.759 (0.740, 0.777) | 0.418 (0.386, 0.454) |

Table B3: **Unimodal performance results for the in-hospital mortality task on the validation set.** Evaluation of unimodal and multimodal models per modality on the multimodal validation set, including the number of training samples per model.

| MODEL | TRAINING SET | CHEST X-RAY IMAGES | | CLINICAL TIME SERIES | |
|---|---|---|---|---|---|
| | | AUROC | AUPRC | AUROC | AUPRC |
| **Multimodal** | | | | | |
| MedFuse-3H | 4,885 | 0.691 (0.629, 0.749) | 0.257 (0.189, 0.356) | 0.828 (0.775, 0.876) | 0.445 (0.338, 0.580) |
| MIND (Ours) | 4,885 | 0.712 (0.650, 0.775) | 0.277 (0.208, 0.375) | 0.858 (0.811, 0.898) | 0.548 (0.432, 0.678) |
| **Unimodal** | | | | | |
| ResNet-34 | 124,671 | 0.732 (0.715, 0.749) | 0.183 (0.167, 0.205) | - | - |
| ResNet-10 | 124,671 | 0.725 (0.709, 0.742) | 0.177 (0.160, 0.196) | - | - |
| 2-layer LSTM | 42,628 | - | - | 0.870 (0.846, 0.891) | 0.528 (0.467, 0.595) |
| 4-layer LSTM | 42,628 | - | - | 0.870 (0.846, 0.892) | 0.536 (0.473, 0.601) |

## B.4 Ablation studies

Table B4 presents a sensitivity analysis (Ablation study I, Section 5.3) for the in-hospital mortality task on the test set, focusing on different loss components. Each setting indicates the components used in the modified loss for training and the type of knowledge distillation performed. Similarly, Table B5 provides the results on the validation set for Ablation Study I related to the prediction of clinical conditions, while Table B6 presents results for the in-hospital mortality prediction task. Table B7 specifies the hyper-parameter values used in Ablation Study II (Section 5.4), which is depicted in Figure 3.

Table B4: **Ablation study on MIND for multimodal and unimodal performance in the in-hospital mortality prediction task**. Fusion and unimodal performance of MIND on the multimodal test set, focusing on different loss components.

| # | $\mathcal{L}_{S_{AB}}$ | $\mathcal{L}_{S_{A/B}}$ | $\mathcal{L}_{KD^U_{A/B}}$ | $\mathcal{L}_{EKD^U_{A/B}}$ | $\omega_{A/B}$ | Fusion | | Chest X-Ray | | Time Series | |
|---|---|---|---|---|---|---|---|---|---|---|---|
| | | | | | | AUROC | AUPRC | AUROC | AUPRC | AUROC | AUPRC |
| 1 | ✓ | | | | | 0.816 (0.785, 0.847) | 0.468 (0.398, 0.546) | - | - | - | - |
| 2 | ✓ | ✓ | | | | 0.820 (0.787, 0.851) | 0.461 (0.392, 0.541) | 0.669 (0.626, 0.713) | 0.276 (0.232, 0.339) | 0.813 (0.783, 0.843) | 0.442 (0.372, 0.523) |
| 3 | ✓ | ✓ | ✓ | | | 0.823 (0.792, 0.852) | 0.454 (0.389, 0.537) | 0.661 (0.618, 0.704) | 0.268 (0.219, 0.333) | 0.818 (0.788, 0.847) | 0.455 (0.381, 0.532) |
| 4 | ✓ | ✓ | | ✓ | | 0.832 (0.803, 0.860) | 0.482 (0.412, 0.563) | 0.661 (0.613, 0.705) | 0.279 (0.232, 0.348) | 0.827 (0.794, 0.857) | 0.478 (0.402, 0.558) |
| 5 | ✓ | ✓ | ✓ | | ✓ | 0.835 (0.804, 0.864) | 0.494 (0.421, 0.574) | 0.682 (0.641, 0.721) | 0.289 (0.237, 0.359) | 0.827 (0.797, 0.856) | 0.484 (0.410, 0.560) |
| 6 | ✓ | ✓ | | ✓ | ✓ | 0.844 (0.815, 0.872) | 0.505 (0.433, 0.587) | 0.689 (0.652, 0.726) | 0.290 (0.233, 0.354) | 0.830 (0.801, 0.859) | 0.502 (0.426, 0.577) |

Table B5: **Ablation study on MIND for multimodal and unimodal performance in the clinical conditions task**. Fusion and unimodal performance of MIND on the multimodal validation set, focusing on different loss components.

| # | $\mathcal{L}_{S_{AB}}$ | $\mathcal{L}_{S_{A/B}}$ | $\mathcal{L}_{KD^U_{A/B}}$ | $\mathcal{L}_{EKD^U_{A/B}}$ | $\omega_{A/B}$ | Fusion | | Chest X-Ray | | Time Series | |
|---|---|---|---|---|---|---|---|---|---|---|---|
| | | | | | | AUROC | AUPRC | AUROC | AUPRC | AUROC | AUPRC |
| 1 | ✓ | | | | | 0.755 (0.716, 0.794) | 0.454 (0.390, 0.529) | - | - | - | - |
| 2 | ✓ | ✓ | | | | 0.757 (0.715, 0.796) | 0.468 (0.399, 0.546) | 0.677 (0.629, 0.722) | 0.370 (0.316, 0.438) | 0.718 (0.674, 0.760) | 0.417 (0.356, 0.489) |
| 3 | ✓ | ✓ | ✓ | | | 0.773 (0.734, 0.811) | 0.486 (0.416, 0.564) | 0.695 (0.648, 0.740) | 0.391 (0.333, 0.461) | 0.739 (0.698, 0.778) | 0.435 (0.371, 0.511) |
| 4 | ✓ | ✓ | | ✓ | | 0.779 (0.738, 0.816) | 0.489 (0.421, 0.567) | 0.699 (0.652, 0.744) | 0.397 (0.337, 0.468) | 0.742 (0.700, 0.781) | 0.437 (0.375, 0.510) |
| 5 | ✓ | ✓ | ✓ | | ✓ | 0.784 (0.745, 0.820) | 0.502 (0.432, 0.580) | 0.700 (0.654, 0.745) | 0.400 (0.341, 0.470) | 0.747 (0.707, 0.786) | 0.445 (0.380, 0.520) |
| 6 | ✓ | ✓ | | ✓ | ✓ | 0.790 (0.752, 0.826) | 0.516 (0.446, 0.593) | 0.710 (0.665, 0.754) | 0.412 (0.353, 0.484) | 0.754 (0.714, 0.792) | 0.459 (0.394, 0.534) |

# C Additional results on multimodal multiclass benchmark datasets

## C.1 Datasets

**CREMA-D** (Cao et al., 2014) is an audio-visual dataset designed for speech emotion recognition. It includes 7,442 video clips from 91 actors, each speaking a selection of 12 sentences. The utterances express, with

Table B6: **Ablation study on MIND for multimodal and unimodal performance in the in-hospital mortality task**. Fusion and unimodal performance of MIND on the multimodal validation set, focusing on different loss components.

| # | $\mathcal{L}_{S_{AB}}$ | $\mathcal{L}_{S_{A/B}}$ | $\mathcal{L}_{KD^U_{A/B}}$ | $\mathcal{L}_{EKD^U_{A/B}}$ | $\omega_{A/B}$ | Fusion | | Chest X-Ray | | Time Series | |
|---|---|---|---|---|---|---|---|---|---|---|---|
| | | | | | | AUROC | AUPRC | AUROC | AUPRC | AUROC | AUPRC |
| 1 | ✓ | | | | | 0.833 (0.778, 0.883) | 0.503 (0.395, 0.619) | - | - | - | - |
| 2 | ✓ | ✓ | | | | 0.836 (0.783, 0.885) | 0.478 (0.388, 0.606) | 0.691 (0.629, 0.749) | 0.257 (0.189, 0.356) | 0.828 0.775, 0.876) | 0.445 (0.338, 0.580) |
| 3 | ✓ | ✓ | ✓ | | | 0.839 (0.784, 0.887) | 0.467 (0.355, 0.606) | 0.670 (0.599, 0.735) | 0.248 (0.187, 0.343) | 0.832 (0.779, 0.881) | 0.467 (0.363, 0.602) |
| 4 | ✓ | ✓ | | ✓ | | 0.842 (0.790, 0.893) | 0.488 (0.381, 0.615) | 0.678 (0.605, 0.745) | 0.280 (0.201, 0.395) | 0.837 (0.786, 0.880) | 0.475 (0.372, 0.605) |
| 5 | ✓ | ✓ | ✓ | | ✓ | 0.858 v(0.810, 0.901) | 0.545 (0.428, 0.674) | 0.698 (0.636, 0.760) | 0.278 (0.205, 0.388) | 0.850 (0.802, 0.892) | 0.543 (0.423, 0.665) |
| 6 | ✓ | ✓ | | ✓ | ✓ | 0.866 (0.817, 0.908) | 0.572 (0.455, 0.692) | 0.712 (0.650, 0.775) | 0.280 (0.211, 0.378) | 0.858 (0.811, 0.898) | 0.548 (0.432, 0.678) |

Table B7: **Sensitivity analysis of the weighting parameters $\omega_A$ and $\omega_B$.** Values of the hyper-parameters used for each setting in Ablation Study II (Section 5.4), where $\omega_A = \omega_{cxr}$ and $\omega_B = \omega_{ehr}$. The corresponding learning curves are depicted in Figure 3.

| Setting | $\omega_{cxr}$ | $\omega_{ehr}$ |
|---|---|---|
| 1. $\omega_{cxr} = \omega_{ehr} = 0$ | 0 | 0 |
| 2. $\omega_{cxr} \ll \omega_{ehr}$ | 1 | 500 |
| 3. $\omega_{cxr} < \omega_{ehr}$ | 10 | 100 |
| 4. $\omega_{cxr} = \omega_{ehr}$ | 10 | 10 |
| 5. $\omega_{cxr} > \omega_{ehr}$ | 100 | 10 |
| 6. $\omega_{cxr} \gg \omega_{ehr}$ | 500 | 1 |

varying degrees of intensity, one of six common emotions: anger, happiness, disgust, fear, neutral, and sadness. The training set comprises 5,210 samples, while both the validation and test sets contain 1,116 samples each.

**S-MNIST** (Khacef et al., 2019) is an audio-visual dataset designed for multimodal fusion classification. It was created by pairing the original MNIST handwritten digits database with a spoken digits database extracted from Google Speech Commands. To construct the training, validation, and test sets, we randomly sampled 8,000 instances for the training set and 2,000 instances for the validation set from the original training dataset (10 classes, grayscale digits 0-9). Additionally, we randomly sampled 2,000 instances from the original test set to create our test set.

**LUMA** (Bezirganyan et al., 2024) is a multimodal dataset designed for benchmarking multimodal learning. The image modality includes images from a 50-class subset of the CIFAR-10 and CIFAR-100 datasets, while the audio modality contains utterances of the class labels. The dataset is imbalanced, with the most prevalent 22 classes having 1,500 paired samples or more. To construct our dataset, we generate an evenly distributed sample, containing 1,500 paired instances per class. The resulting dataset, which includes 33,000 paired instances, is randomly split into training, validation, and test sets, with 25,000, 4,000, and 4,000 audio-video samples, respectively.

## C.2 Architectures and Implementation

We utilize a multimodal architecture composed of ResNet variants as modality encoders for both modalities. The outputs of the modality encoders are concatenated and passed through a linear layer that computes the final prediction. Specifically, the details of the vanilla multimodal architectures and their sizes are provided as follows:

- For the S-MNIST dataset, we use ResNet-10 models, trained on the entire training set, as unimodal teachers and ResNet-3 models as modality encoders for the compressed model, which is trained with a fraction of the original dataset (8,000 randomly selected samples). The teacher models have approximately 4.9 million parameters, while the whole MIND model has only approximately 151,000 parameters (32.5x size reduction).

- For the CREMA-D dataset, we use ResNet-18 models as unimodal teachers and ResNet-6 models as modality encoders for the compressed model. All models are trained on the full training dataset. The teacher models have approximately 11 million parameters, while the whole MIND model has only approximately 612,000 parameters (18x size reduction).

- For the LUMA dataset, we use ResNet-10 models as unimodal teachers and ResNet-3 models as modality encoders for the compressed model. All models are trained on the entire training dataset. The teacher models have approximately 4.9 million parameters, while the whole MIND model has only approximately 151,000 parameters (32.5x size reduction).

For the MIND model and all baselines, we perform random hyper-parameter tuning of the learning rates and weighting coefficients where applicable. We conduct a minimum of 50 runs per model for each task. For the unimodal baselines, we randomly select ten different learning rates and use the best-performing model based on the epoch with the highest classification accuracy on the validation set. In general, we use a batch size of 64 and train the models for 50 to 100 epochs. The learning rates used are within the ranges described in Table A3.

