# OpenReview forum: "MIND: Modality-Informed Knowledge Distillation Framework for Multimodal Clinical Prediction Tasks"
_TMLR — Accepted by TMLR_

### Review · Reviewer_zzZQ · 2024-10-18

**Summary Of Contributions:**

This paper proposes a Modality-INformed knowledge Distillation (MIND) framework to leverage knowledge distillation to transfer knowledge from ensembles of multiple pre-trained deep neural networks of varying sizes into a smaller multimodal student. Experimental results show that the proposed method can achieve performance comparable to that of other SOTA methods on various datasets.

**Audience:**

Yes

**Broader Impact Concerns:**

There are no concerns on the ethical implications.

**Claims And Evidence:**

No

**Requested Changes:**

Please address the questions in the Cons and revise the manuscript accordingly.

**Strengths And Weaknesses:**

**Pros**:
+ The proposed weighted unimodal ensemble knowledge distillation loss may interest some TMLR audiences. This loss can be used to compress models for multimodal tasks.
+ The ablation study can generally verify the effectiveness of the key components in the proposed method.
+ The overall performance is comparable to the other SOTA methods on various tasks/datasets.

**Cons**:

Overall, this paper is hard to follow, with some statements not well clarified or experimentally supported.

1. Why do unimodal teacher networks allow the student to learn diverse representations? How about if the teacher model is not an unimodal network? And what is the unimodal model? Does it mean modal-specific models?

2. Though the proposed method is claimed to improve the compression of multimodal networks, it is unclear whether the performance of the compressed networks owes to the extra large-scale datasets, compared with the other SOTA methods (see Q8). Regarding efficiency, the proposed method needs to train multiple teacher networks for each modality during the training stage, so it can be limited to only a few modalities and can be computationally expensive when there are many different modalities, e.g., MRI, PET, and Ultrasound in addition to X-ray.

3. The proposed method looks like a two-step method, with the first step training various teacher models and the second step training a student model using the ensemble of teachers. A two-step method can need more training time and multiple teachers pre-trained on large-scale datasets can cost more computational resources. Due to the cumbersome pre-training of the teachers in the first step, one may consider using some other methods, e.g., MedFuse, for the first step training to save time/resources. That is, what will happen if other SOTA methods are used as the teacher model?

4. On page 4, why are the randomly initialized encoders more comparable in the proposed setup (i.e., joint fusion, I suppose)? Particularly, why is the initialization scheme related to the setup?

5. Though Sec. 5.4 analyzes balancing multimodal learning with \omiga_A and \omiga_A, it does not clearly show how to set \omiga_A and \omiga_A in Eq. (7)? What are the exact values of these two hyper-parameters during the training phase?

6. The number of teachers for each modality is 3, but why 3 teachers are chosen? Is 3 the best?

7. How is Eq. (6) used during training (except for plotting Fig. 3)? That is, when Fig. 3 is plotted, how to adjust the training hyperparameters/settings according to the Figure?

8. Whether the CXR dataset and the EHR dataset used for training unimodal models are the extra data in addition to the datasets used for clinical conditions and in-hospital tasks? Whether the other competitors like MedFuse use the same amount of data as the CXR and EHR datasets or not?

9. In Sec. 6, “Compared to existing work, only Bucilua et al. (2006) combine ensemble learning and knowledge distillation to train compact networks.” This statement seems to be incorrect. For instance, [A] also combines ensemble learning and knowledge distillation to train compact networks. Please conduct a comprehensive literature review to better highlight the difference between the proposed method and the others.

[A] Fukuda, Takashi, et al. "Efficient Knowledge Distillation from an Ensemble of Teachers." Interspeech. 2017.

---

> ### Author Response · Authors · 2024-11-09
> **Authors’ Response to Review of Paper 3419 by Reviewer zzZQ**
>
> The authors would like to thank the reviewer for the thorough review and for highlighting relevant aspects. Below, we provide a point-by-point response to the questions raised in the comments, along with corresponding updates to the revised manuscript:
>
> > 1. Overall, this paper is hard to follow, with some statements not well clarified or experimentally supported.
>
> Answer: We agree with the reviewer that some parts of the original submission could benefit from a revision. In the updated manuscript, we clarified our claims and removed those without clear experimental support. We also proofread the whole paper to improve readability. We highlighted all these modifications in blue in the revised version of the paper.
>
> > 2. Why do unimodal teacher networks allow the student to learn diverse representations? How about if the teacher model is not an unimodal network? And what is the unimodal model? Does it mean modal-specific models?
>
> Answer: We thank the reviewer for their attention to detail.
> - We agree with the reviewer in that the correct statement should be ‘allow the student to learn **from** diverse representations’ (that is, from the unimodal teacher ensembles). We have corrected this in the abstract in the revised manuscript.
> - In the development of our proposal, we initially included an ensemble of larger multimodal teachers. However, based on our experiments, their contribution did not lead to an improvement in performance. For that purpose, our method proposes leveraging strong unimodal teachers, which provide a significant boost in performance, and not multimodal teachers. In addition, this lowers the computational cost of pre-training by excluding the pre-training of multimodal teachers from our framework.
> - As the reviewer points out, a unimodal model is a modality-specific model. They are models trained with single modality data samples (e.g., CXR). To further clarify this point, we added the following text in page 1 (Introduction, end of first paragraph, highlighted in blue): *Note that the term unimodal network refers to a modality-specific model, i.e., a  model trained with samples containing a single data modality (e.g., image), while the term multimodal model refers to a network trained using samples including multiple data modalities*.
>
> > 3. Though the proposed method is claimed to improve the compression of multimodal networks, it is unclear whether the performance of the compressed networks owes to the extra large-scale datasets, compared with the other SOTA methods (see Q8).
>
> Answer: The authors would like to point out that due to the inherent limitations of multimodal datasets such as data availability, they are typically smaller in size than unimodal datasets. Addressing the comment of the reviewer, all SOTA methods (i.e., TS [1], MKE [2] and UME [3]) used for comparison with the MIND framework use larger dataset sizes for the unimodal models. For instance, this fact is explicitly stated in [1], where the authors clarify that ‘*since the teacher models were trained on relatively larger datasets compared with the datasets used to train the student model, the teachers were experts on each modality and the expertise could help the student to improve the performance*’. Following the same principle, the MIND framework uses larger datasets for the unimodal teachers, and smaller datasets for the multimodal student. In this comparable setting, as our experiments demonstrate, the MIND framework outperforms all SOTA methods across all datasets, tasks and fusion methods.
>
> [1] Qi Wang, Liang Zhan, Paul Thompson, and Jiayu Zhou. Multimodal learning with incomplete modalities by knowledge distillation. In Proceedings of the 26th ACM SIGKDD International Conference on Knowledge Discovery & Data Mining, pp. 1828–1838, 2020.
>
> [2] Zihui Xue, Sucheng Ren, Zhengqi Gao, and Hang Zhao. Multimodal knowledge expansion. In Proceedings of the IEEE/CVF International Conference on Computer Vision, pp. 854–863, 2021.
>
> [3] Chenzhuang Du, Jiaye Teng, Tingle Li, Yichen Liu, Tianyuan Yuan, Yue Wang, Yang Yuan, and Hang Zhao. On uni-modal feature learning in supervised multi-modal learning. In International Conference on Machine Learning, pp. 8632–8656. PMLR, 2023.

---

> ### Author Response · Authors · 2024-11-09
> **Authors’ Response to Review of Paper 3419 by Reviewer zzZQ**
>
> > 4. Regarding efficiency, the proposed method needs to train multiple teacher networks for each modality during the training stage, so it can be limited to only a few modalities and can be computationally expensive when there are many different modalities, e.g., MRI, PET, and Ultrasound in addition to X-ray.
>
> Answer: We agree with the reviewer that multiple teacher networks have to be trained for each modality as a prior step and to generate the teacher ensembles. In this regard, we would like to note that the implementation of offline Knowledge Distillation (KD), the KD paradigm embedded into the MIND and all SOTA methods evaluated in our work, require the training of multiple teacher networks as a prior step for teacher-student knowledge distillation. In this setting, multiple pre-trained teacher models are trained and, typically, the best pre-trained unimodal network is used as a teacher model for offline knowledge distillation, that is, to generate outputs used for student training. Offline KD therefore, always involves this two-step process.
>
> We agree with the reviewer that the first step, teacher training, can be computationally expensive, especially for a large number of modalities. In this regard, it is worth emphasizing that most multimodal work combines two modalities, rarely surpassing more than 3 modalities. In any case, this is an inherent limitation of offline KD, and, consequently of our work, which focuses on student model knowledge compression (in terms of performance and size) rather than overall training and computational efficiency. However, the MIND framework proposes a step forward by ensembling all these pre-trained unimodal teacher models (instead of discarding them and selecting the best one) for improved multimodal and unimodal performance. For these cases and with a moderate number of teachers, the MIND framework can provide a good tradeoff between model efficacy and training efficiency.
>
> > 5. The proposed method looks like a two-step method, with the first step training various teacher models and the second step training a student model using the ensemble of teachers. A two-step method can need more training time and multiple teachers pre-trained on large-scale datasets can cost more computational resources. Due to the cumbersome pre-training of the teachers in the first step, one may consider using some other methods, e.g., MedFuse, for the first step training to save time/resources. That is, what will happen if other SOTA methods are used as the teacher model?
>
> Answer: The MIND framework is embedded within the offline KD paradigm. The current SOTA in this KD paradigm establishes a two-step method, as the reviewer describes, involving first the training of various teacher models and, a second step for training the student model. This same two-step procedure is used in all SOTA methods involving offline KD with which MIND is compared (e.g., TS [1], MKE [2], UME [3]). While we agree that this two-step process requires more computational resources, it is one of the essential components of offline KD. Addressing this increase in computational resources, a distinctive feature of MIND is that we propose to ensemble them instead of discarding sub-optimal teacher models, thus lowering the computational waste.
>
> Our framework proposes an ensemble of unimodal teachers as one of its distinctive features. However, aligning with the reviewer suggestion, our initial proposal included an additional multimodal KD component (i.e., distillation from multimodal teacher). Thus, in the design of the framework we considered the addition of multimodal teachers. Notwithstanding that, the experimental results did not provide any additional benefit from the multimodal distillation on top of the unimodal distillation. Therefore, while the idea suggested by the reviewer is coherent and enticing, our experimental results did not justify the addition of a multimodal knowledge distillation component to our proposed framework.
>
> [1] Qi Wang, Liang Zhan, Paul Thompson, and Jiayu Zhou. Multimodal learning with incomplete modalities by knowledge distillation. In Proceedings of the 26th ACM SIGKDD International Conference on Knowledge Discovery & Data Mining, pp. 1828–1838, 2020.
>
> [2] Zihui Xue, Sucheng Ren, Zhengqi Gao, and Hang Zhao. Multimodal knowledge expansion. In Proceedings of the IEEE/CVF International Conference on Computer Vision, pp. 854–863, 2021.
>
> [3] Chenzhuang Du, Jiaye Teng, Tingle Li, Yichen Liu, Tianyuan Yuan, Yue Wang, Yang Yuan, and Hang Zhao. On uni-modal feature learning in supervised multi-modal learning. In International Conference on Machine Learning, pp. 8632–8656. PMLR, 2023.

---

> ### Author Response · Authors · 2024-11-09
> **Authors’ Response to Review of Paper 3419 by Reviewer zzZQ**
>
> > 6. On page 4, why are the randomly initialized encoders more comparable in the proposed setup (i.e., joint fusion, I suppose)? Particularly, why is the initialization scheme related to the setup?
>
> Answer: We agree with the reviewer that this comment (located in the first line of page 4) is not in the right location within the text. It is actually related to the experimental evaluation and comparison of approaches, not a proposal related to our framework. What we meant here is that all baselines including our framework are initialized randomly for a fair comparison across methods (i.e., our proposed approach and SOTA methods). We have moved this comment to Section 4.Experiments (Baselines). We have highlighted the relocated text in blue.
>
> > 7. Though Sec. 5.4 analyzes balancing multimodal learning with \omiga_A and \omiga_A, it does not clearly show how to set \omiga_A and \omiga_A in Eq. (7)? What are the exact values of these two hyper-parameters during the training phase?
>
> Answer: We agree with the reviewer that while Section 5.4 demonstrates how omega_A​ and omega_B​ can be used to balance multimodal learning, the original version of the manuscript did not provide sufficient information regarding the hyperparameter values. Additionally, the use of comparison operators (e.g., <<, >) was intended to simplify the understanding of the impact of these hyperparameters by illustrating extreme values; however, this approach can be considered vague. Addressing this comment, we have provided the exact values of these two-hyperparameters during the training phase for each setting in Appendix B.4. We have added a new table providing this information (Table B7 in the revised manuscript, highlighted in blue). We have added the corresponding reference in the main text and some explanatory text in the corresponding appendix.
>
> The values of omega_A and omega_B should be considered as hyper-parameters and, consequently, should be tuned for any given task / dataset. However, based on our empirical investigation, we can recommend starting with larger weights for the under-utilized modality and increasing or decreasing according to the main goal. For instance, aiming for balancing multimodal learning can create stronger encoders, but it may impact the multimodal performance. However, if multimodal performance is the main objective, overfitting towards the strongest modality (i.e., larger weight) may yield enhanced performance.
>
> > 8. The number of teachers for each modality is 3, but why 3 teachers are chosen? Is 3 the best?
>
> Answer: We appreciate the attention to detail from the reviewer. The ensemble learning literature does not provide any clear consensus on the optimal ensemble size. For instance, in [1] the ensemble size of 4 provided optimal results, while in [2] the optimal sizes varied significantly from 3 to 50 members depending on the regression task and dataset employed. In [3], the authors argue that for classification, the optimal ensemble size is equal to the number of classes. Closer work to ours like [4] and [5], used an ensemble of 2 teachers for their experiments. In our work, as a compromise between computational cost and performance improvement, we selected an ensemble size of 3 for our experiments, showcasing the benefits of ensembling in the framework. We also preferred to keep the number of members per ensemble low, to keep the computational cost of the pre-training stage at a minimum. We preferred to keep the ensemble size fixed for all tasks, regardless of the task (binary, multilabel, and multiclass), as a trade-off between computational efficiency, and model efficacy. While we do not discard that larger ensembles may provide better improvement for particular tasks, our results showcase that an ensemble of 3 can provide a good performance boost compared to a single teacher (see Table 4 and Table B4), showcasing the benefits of the proposed framework across tasks and datasets. In future work, we plan to comprehensively investigate the impact of ensemble size on model performance, which is currently beyond the scope of the current study. In this work, we limited our exploration to prove that the ensemble improves the performance with respect to a single teacher.
>
> [1] Zhang, Xiaohui, et al. "A diversity-penalizing ensemble training method for deep learning." INTERSPEECH. 2015.
>
> [2] Milinski, Sebastian, et al. "How large does a large ensemble need to be?." Earth System Dynamics 11.4 (2020).
>
> [3] Bonab, Hamed, and Fazli Can. "Less is more: A comprehensive framework for the number of components of ensemble classifiers." IEEE Transactions on neural networks and learning systems 30.9 (2019).
>
> [4] Fukuda, Takashi, et al. "Efficient Knowledge Distillation from an Ensemble of Teachers." Interspeech. 2017.
>
> [5] Li, Xingjian, et al. "“In-network ensemble”: Deep ensemble learning with diversified knowledge distillation." ACM Transactions on Intelligent Systems and Technology (TIST) 12.5 (2021).

---

> ### Author Response · Authors · 2024-11-09
> **Authors’ Response to Review of Paper 3419 by Reviewer zzZQ**
>
> > 9. How is Eq. (6) used during training (except for plotting Fig. 3)? That is, when Fig. 3 is plotted, how to adjust the training hyperparameters/settings according to the Figure?
>
> Answer: We use Eq. (6) to showcase modality overfitting during training. However, our methodology does not actually use Eq. (6) during training. We believe that using Eq. (6) and plotting training curves using different training hyperparameters such as in Fig. 3, allows us to demonstrate that our proposed methodology can be leveraged to balance multimodal learning. However, this is a by-product of our framework, an additional benefit and potential use of our proposed methodology, which focuses on enhancing multimodal and unimodal performance for compressed models.
>
> > 10. Whether the CXR dataset and the EHR dataset used for training unimodal models are the extra data in addition to the datasets used for clinical conditions and in-hospital tasks? Whether the other competitors like MedFuse use the same amount of data as the CXR and EHR datasets or not?
>
> Answer: We appreciate the comment of the reviewer. We would like to point out that to ensure a fair comparison across methods, all multimodal models, regardless of the method used, use the same datasets and all unimodal models, regardless of the method used, use the same dataset sizes for each respective modality. Unimodal datasets are typically larger in size compared to the multimodal datasets, which requires the pairing of data modalities to construct a data sample. To further clarify this point in the text, we have modified Table A2 to incorporate the sizes of each unimodal and multimodal dataset used. We have also added a clarification statement in Section 4 (Baselines) stating that ‘’*All MIND and baseline models trained for each specific task are all trained using the same dataset sizes, as described in Appendix A.2 and Appendix C.1.*”. The changes are highlighted in blue in the revised manuscript.
>
> > 11. In Sec. 6, “Compared to existing work, only Bucilua et al. (2006) combine ensemble learning and knowledge distillation to train compact networks.” This statement seems to be incorrect. For instance, [A] also combines ensemble learning and knowledge distillation to train compact networks. Please conduct a comprehensive literature review to better highlight the difference between the proposed method and the others.
>
> Answer: We appreciate the comment of the reviewer. We agree with the reviewer that we are not the first approach proposing ensemble learning and knowledge distillation. We have addressed the issue and improved the paragraph, which now reads as follows:
>
> *While Bucilua et al. (2006), Fukuda et al. (2017), Asif et al. (2020), Li et al. (2021), and Wu et al. (2022a) are examples of previous work combining ensemble learning and offline response-based knowledge distillation to train compact networks, these approaches focus their work on unimodal networks. In contrast, we are concerned with the particularities of multimodal joint fusion networks, both in performance and size, thus we propose incorporating and weighting modality-specific ensembles during student knowledge distillation as a distinctive feature. Despite methodological differences, MIND similarly avoids the need for a large ensemble of classifiers, which demands significant resources during inference. In addition, the MIND framework can be used to address imbalanced multimodal learning during training and leverages modality encoders to handle unimodal samples, achieving unimodal performance on par with powerful unimodal models.*
>
> New references added to the revised version:
>
> Fukuda, Takashi, et al. "Efficient Knowledge Distillation from an Ensemble of Teachers." Interspeech. 2017.
>
> Wu, Chuhan, et al. "Unified and effective ensemble knowledge distillation." arXiv preprint arXiv:2204.00548 (2022).
>
> Asif, Umar, Jianbin Tang, and Stefan Harrer. "Ensemble knowledge distillation for learning improved and efficient networks." ECAI 2020. IOS Press, 2020. 953-960.
>
> Li, Xingjian, et al. "“In-network ensemble”: Deep ensemble learning with diversified knowledge distillation." ACM Transactions on Intelligent Systems and Technology (TIST) 12.5 (2021): 1-19.

---

> > ### Comment · Reviewer_zzZQ · 2024-11-16
> > **Some concerns were addressed, but some not**
> >
> > Thanks for the response which has addressed some of my concerns. However, the response also raises some new confusion:
> >
> > The answer to Q2: “we initially included an ensemble of larger multimodal teachers”. What kind of method is used for the ensemble, and what are the results for the multimodal teachers and the “strong unimodal teachers”? More details should be provided to make the statements more convincing.
> >
> > For Q4, “most multimodal work combines two modalities, rarely surpassing more than 3 modalities. In any case, this is an inherent limitation of offline KD ...” The concept of “offline KD” has not been introduced in the manuscript before. If this is an inherent limitation of offline KD, please introduce the offline KD in the manuscript and discuss this limitation of the proposed method.
> >
> > For Q5, “our initial proposal included an additional multimodal KD component...the experimental results did not provide any additional benefit”. Again, what is the “additional multimodal KD component”? What are their experimental results?
> >
> > For Q6, “We have moved this comment to Section 4. Experiments (Baselines)” and the text in Section 4 is that “We believe that learning with randomly initialized encoders is more comparable in the proposed setup.” Why are the randomly initialized encoders more comparable in the proposed setup rather than following previous work (fine-tuning pre-trained unimodal encoders)?
> >
> > For Q8, “our results showcase that an ensemble of 3 can provide a good performance boost compared to a single teacher (see Table 4 and Table B4), showcasing the benefits of the proposed framework across tasks and datasets.” Sorry that I did not find any results comparing the ensemble of 3 with the other number of model ensembles. Please clearly point out which results are of 3 ensembles, and which are of 1, 2 (or 4) ensembles.
> >
> > For Q9, “our methodology does not actually use Eq. (6) during training”. It is misleading to include Eq. (6) between Eq. (5) and (7) as it causes confusion on how it is related to the training objectives in Eq. (5,7). If it is not used during training but only used as an analytic tool, Eq. (6) can be introduced in other positions.
> >
> > For Q11, “... leverages modality encoders to handle unimodal samples, achieving unimodal performance on par with powerful unimodal models.” What are “unimodal samples” (this concept also appears in the revised Abstract as well as many other sections in the revision)? What is “unimodal performance”?

---

> ### Author Response · Authors · 2024-11-19
> **Authors’ Response to 'Some concerns were addressed, but some not' by Reviewer zzZQ - part 1**
>
> The authors would like to thank the reviewer for the thorough review and for highlighting relevant aspects. Below, we provide a point-by-point response to the questions raised in the comments, along with corresponding updates to the revised manuscript:
>
> > The answer to Q2: “we initially included an ensemble of larger multimodal teachers”. What kind of method is used for the ensemble, and what are the results for the multimodal teachers and the “strong unimodal teachers”? More details should be provided to make the statements more convincing.
>
> > For Q5, “our initial proposal included an additional multimodal KD component...the experimental results did not provide any additional benefit”. Again, what is the “additional multimodal KD component”? What are their experimental results?
>
> Answer: As we explained in our previous response, we experimented with an  ensemble of multimodal models as teachers  of a smaller multimodal student, which represents traditional knowledge distillation from a large teacher to a smaller student (in terms of number of parameters). The multimodal ensemble consisted of a larger multimodal teacher (for the multimodal prediction) and two unimodal teachers(one for each encoder).In our preliminary results, as shown below, we observed that using the ensemble with the multimodal teacher  yielded worse performance compared to using unimodal teachers only, i.e., our MIND framework. We believe this may be due to the same limitation that the multimodal teachers are learning from smaller datasets. Hence, we focused our proposal on modality distillation and did not include not including multimodal distillation. This also saves computational cost, as highlighted by the reviewer in their previous question, as it alleviates the need of training another set of teachers. The performance of the strong unimodal encoders is provided in Table 2 and Table 3, i.e., unimodal models listed as Resnet-34, ResNet-10, 2-layer LSTM, and 4-layer LSTM. As we can observe, MIND performs on par with these larger models that are trained with significantly more data.
>
> | Ensemble Distillation                                                                                                   | Multimodal AUROC | Multimodal AUPRC | CXR AUROC | CXR AUPRC | EHR AUROC | EHR AUPRC |
> |-------------------------------------------------------------------------------------------------------------------------|-------|-------|-----------|-----------|-----------|-----------|
> | Ensemble of two modality-specific teachers and one multimodal teacher                                                   | 0.776 | 0.490 | 0.694     | 0.390     | 0.739     | 0.430     |
> | Ensemble excluding multimodal teacher (unimodal teachers only) = Our MIND framework(extracted from Table 2 and Table 3) | **0.790** | **0.516** | **0.710**    | **0.412**     | **0.754**     | **0.459**     |
>
> > For Q4, “most multimodal work combines two modalities, rarely surpassing more than 3 modalities. In any case, this is an inherent limitation of offline KD ...” The concept of “offline KD” has not been introduced in the manuscript before. If this is an inherent limitation of offline KD, please introduce the offline KD in the manuscript and discuss this limitation of the proposed method.
>
> Answer: We introduce the offline KD setting at the end of page 2 (Background Information section, Knowledge Distillation). Specifically, the paragraph says: ‘*In **this offline setting**, the aim of the student network is to mimic the performance of the teacher network by approaching the softened responses of the pre-trained teacher network.*‘ We agree that it might be a subtle mention. For that purpose we have modified the statement to explicitly state ‘*In the **offline knowledge distillation setting**, the aim of the student network is to mimic the performance of the teacher network by approaching the softened responses of the pre-trained teacher network*.’ In addition, we have added a paragraph in the Limitations section (page 13) stating that ‘*Specifically, our framework is embedded within the offline KD paradigm. The current state of the art in this KD paradigm establishes a two-step method: first, the teacher models are trained, and then the student model is trained by distilling information from the teachers. This two-step process is essential to offline KD, which requires significant computational resources. However, to address this increase in resource demand, a distinctive feature of MIND is our proposal to ensemble the teacher models instead of discarding sub-optimal ones, thereby reducing computational waste*’. The added fragments have been highlighted in blue in the revised version of the manuscript.

---

> ### Author Response · Authors · 2024-11-19
> **Authors’ Response to 'Some concerns were addressed, but some not' by Reviewer zzZQ - part 2**
>
> > For Q6, “We have moved this comment to Section 4. Experiments (Baselines)” and the text in Section 4 is that “We believe that learning with randomly initialized encoders is more comparable in the proposed setup.” Why are the randomly initialized encoders more comparable in the proposed setup rather than following previous work (fine-tuning pre-trained unimodal encoders)?
>
> Answer: We believe that using randomly initialized encoders is more comparable in the proposed setup for several reasons:
>
> - **Baseline consistency**: Randomly initialized encoders provide a consistent baseline that allows for a fair comparison across different models. This approach ensures that any observed performance improvements can be attributed to the architecture and training methodology rather than the advantages of pre-trained weights.
>
> - **Focus on learning dynamics**: By starting with randomly initialized encoders, we can better isolate the learning dynamics specific to our proposed setup. This contrasts with fine-tuning pre-trained encoders, where the initial conditions and learned representations can heavily influence the results, potentially skewing comparisons.
>
> - **Novelty of the proposed framework**: Our setup introduces unique aspects that may not align well with the assumptions underlying previous work. Random initialization allows us to explore how well the model can adapt from scratch, which is crucial for understanding its capabilities in this new context.
>
> - **Avoiding pre-training bias**: Fine-tuning pre-trained unimodal encoders can introduce biases based on the data and tasks they were originally trained on. By using randomly initialized encoders, we eliminate these biases, allowing for a more objective evaluation of our proposed method.
>
> In summary, randomly initialized encoders help us establish a clearer, more equitable framework for assessing the effectiveness of our approach in the context of the proposed setup.
>
> Addressing the comment of the reviewer, we have summarized the above points and added them to the revised version of the manuscript (pages 6-7). The text now reads as follows:
>
> ‘*We believe that learning with randomly initialized encoders is more appropriate in our proposed setup for several reasons. First, randomly initialized encoders establish a consistent baseline for fair comparisons across models, ensuring that any performance improvements reflect the architecture and training methods rather than pre-trained weights. Additionally, starting with random initialization helps isolate the specific learning dynamics of our setup, avoiding the potential of skewing the performance as a result of using fine-tuned pre-trained encoders. Our approach also introduces unique elements that may not align with previous assumptions, making it essential to assess how well the model adapts from scratch to understand its capabilities fully. Finally, using randomly initialized encoders eliminates biases from original training data associated with fine-tuning, allowing for a more objective evaluation of our method.*’
>
> > For Q8, “our results showcase that an ensemble of 3 can provide a good performance boost compared to a single teacher (see Table 4 and Table B4), showcasing the benefits of the proposed framework across tasks and datasets.” Sorry that I did not find any results comparing the ensemble of 3 with the other number of model ensembles. Please clearly point out which results are of 3 ensembles, and which are of 1, 2 (or 4) ensembles.
>
> Answer: The third row of Table 4 and Table B4 provide the results using a single teacher model (notice the check mark in column L_{KD^U_{A/B}}), while the results using an ensemble instead of a single teacher are provided in the fourth row (notice the check mark in column L_{EKD^U_{A/B}}). We also evaluated the combination of weighting a single teacher and ensemble of 3 teachers. These results are provided in row fifth and sixth, respectively. We would like to note that these settings are thoroughly described in Section 5.3, which describes the experiments as follows:
>
> - 3. Supervised learning with unweighted unimodal single-teacher KD. Specifically, we set ωA, ωB = 1 in Equation 6 and use a single unimodal model as the modality teacher.
> - 4. Supervised learning with weighted unimodal single-teacher KD. We apply Equation 6 using a single unimodal model as the modality teacher.
> - 5. Supervised learning with unweighted unimodal EKD. We set ωA, ωB = 1 in Equation 6 and use an ensemble of three teacher models as teachers.
> - 6. MIND: supervised learning with weighted unimodal EKD (Equation 6).
>
> Addressing the comment of the reviewer, in order to make Table 4 and B4 easier to read and interpret, matching the rows with the description in Section 5.3, we have numbered the rows, aligning the numbered description with the row numbers. We have also made the same changes in Table B5 and Table B6. We have highlighted the changes in blue.

---

> > ### Author Response · Authors · 2024-11-19
> > **Authors’ Response to 'Some concerns were addressed, but some not' by Reviewer zzZQ - part 3**
> >
> > > For Q9, “our methodology does not actually use Eq. (6) during training”. It is misleading to include Eq. (6) between Eq. (5) and (7) as it causes confusion on how it is related to the training objectives in Eq. (5,7). If it is not used during training but only used as an analytic tool, Eq. (6) can be introduced in other positions.
> >
> > Answer: We agree with the reviewer that the previous configuration was somewhat confusing. In response to the reviewer’s comment, we have moved the introduction of the conditional utilization rate to Section 5.4 (Results Section), outside of the proposed methodology. We have integrated it with the existing content in this section, thereby avoiding its introduction alongside the proposed methodology. This adjustment ensures that it does not interfere with the presentation of our proposed loss and, we believe, significantly reduces confusion. We appreciate the reviewer’s attention to detail, which has significantly enhanced the readability and flow of the paper.
> >
> >
> > > For Q11, “... leverages modality encoders to handle unimodal samples, achieving unimodal performance on par with powerful unimodal models.” What are “unimodal samples” (this concept also appears in the revised Abstract as well as many other sections in the revision)? What is “unimodal performance”?
> >
> > Answer: To explain the concept of unimodal samples, we have added the following text at the end of page 4 (Methods. Multimodal loss): Note that the term unimodal sample refers to data samples that contain only one modality (e.g., A or B), whereas multimodal samples involve the presence of more than one modality (e.g., A and B). This is the first mention of unimodal samples in the main text, thus we believe the best point to introduce the concept. The added text has been highlighted in blue.
> >
> > Regarding the term unimodal performance, we agree that it is ambiguous, thus have modified the text to be more precise, adding ‘...achieving **predictive performance for unimodal samples that is** on par with powerful unimodal models.’ The added text has been highlighted in blue.

---

> > > ### Comment · Reviewer_zzZQ · 2024-11-19
> > > **Thanks for the further clarification**
> > >
> > > Many thanks for the clarification. Most of my concerns have been addressed, with minor comments: 1) "Our setup introduces unique aspects that may not align well with the assumptions underlying previous work". It would be better to briefly present the assumption using a few words to facilitate understanding. 2) "Multimodal samples involve the presence of more than one modality". It is still confusing why a sample can have multiple modalities. I believe a sample can be either MRI or X-ray modality, but not both. Please explain why a sample can "involve the presence of more than one modality".

---

> ### Author Response · Authors · 2024-11-20
> **Authors’ Response to 'Thanks for the further clarification' by Reviewer zzZQ**
>
> The authors would like to thank the reviewer for their thorough review and acknowledgment of the clarifications. Below, we provide a point-by-point response to the minor comments pointed out by the reviewer, along with the corresponding updates to the revised manuscript:
>
> > 1) "Our setup introduces unique aspects that may not align well with the assumptions underlying previous work". It would be better to briefly present the assumption using a few words to facilitate understanding.
>
> Answer: We agree with the reviewer that further clarification is needed. In this regard, while our framework makes no assumptions, other proposed frameworks involve the use of pre-trained weights as a first step for their training phase. Addressing the reviewer’s comment we have modified the paragraph as follows (the new text is highlighted in bold):
>
> ‘*We believe that learning with randomly initialized encoders is more appropriate in our proposed setup for several reasons. First, randomly initialized encoders establish a consistent baseline for fair comparisons across models, ensuring that any performance improvements reflect the architecture and training methods rather than pre-trained weights. Additionally, starting with random initialization helps isolate the specific learning dynamics of our setup, avoiding the potential of skewing the performance as a result of using fine-tuned pre-trained encoders. Our approach also introduces unique elements that may not align with **assumptions of previous work, such as avoiding the need for pre-training the student model, so we can assess how well the model learns from scratch to understand its full capabilities**. Finally, using randomly initialized encoders eliminates biases from original training data associated with fine-tuning, allowing for a more objective evaluation of our method.*’
>
>
> > 2) "Multimodal samples involve the presence of more than one modality". It is still confusing why a sample can have multiple modalities. I believe a sample can be either MRI or X-ray modality, but not both. Please explain why a sample can "involve the presence of more than one modality".
>
> Answer: In the context of multimodal data where there could be missing modalities, recent work generally defines “multimodal samples” as those that have all modalities present and “unimodal samples” as those with a single modality due to the missingness of other modalities [1-3]. Hence, for example, for breast cancer diagnosis, a multimodal sample could include both a mammogram and an ultrasound image, while a unimodal sample includes one of the two modalities. In the context of our dataset, multimodal samples have both CXR and EHR, while unimodal samples of EHR contain EHR only and unimodal samples of CXR contain CXR only. We modified the sentence above to say: “Multimodal samples involve the presence of both modalities.”
>
> [1] Zhang, Chaohe, et al. "M3care: Learning with missing modalities in multimodal healthcare data." Proceedings of the 28th ACM SIGKDD Conference on Knowledge Discovery and Data Mining. 2022.
>
> [2] Rani, Samta, et al. "Diagnosis of breast cancer molecular subtypes using machine learning models on unimodal and multimodal datasets." Neural Computing and Applications 35.34 (2023): 24109-24121.
>
> [3] Dong, Xiao, et al. "M5product: Self-harmonized contrastive learning for e-commercial multi-modal pretraining." Proceedings of the IEEE/CVF Conference on Computer Vision and Pattern Recognition. 2022.

---

> > ### Comment · Reviewer_zzZQ · 2024-11-20
> > **No further concerns**
> >
> > Many thanks for the clarification. My concerns have been addressed by the authors.

---

### Review · Reviewer_sy5L · 2024-10-31

**Summary Of Contributions:**

This paper proposes Modality-INformed knowledge distillation (MIND), a (small) multimodal knowledge distillation model for small medical datasets. MIND distills modality-specific knowledge from ensembles of unimodal teacher models to a smaller multimodal student model. It uses an early fusion framework but leverages two additional modality-specific losses to enable independent unimodal modules for missing modalities. In addition, it has weighting parameters to balance the unimodal distillation components. On the clinical dataset MIMIC-CXR and non-clinical datasets CREMA-D, S-MNIST, and LUMA, the model achieves SOTA results on most metrics.

**Audience:**

Yes

**Broader Impact Concerns:**

A broader impact statement on societal issues, such as the potential risks of malicious usage of medical models, as well as implicit biases learned from knowledge distillation, should be added.

**Claims And Evidence:**

Yes

**Requested Changes:**

**Related work**: Could the authors discuss [1], which is very similar to this work, where the knowledge is transferred by learning the teacher’s behavior within each modality, and weighting schemes are also introduced. The following similar related work seemingly lessens the novelty of this work.

[1] Jin, Woojeong, et al. "Msd: Saliency-aware knowledge distillation for multimodal understanding." arXiv preprint arXiv:2101.01881 (2021).

**Claim on missing modality**: Could the authors clarify the first paragraph in 5.1: “while enabling unimodal predictions when modalities are missing”. Experiments in Table 1 show that unimodal distillation helps but did Table 1 explicitly test the missing modality scenario? In general for the paper, the reviewer apologizes in advance if she or he missed it, but is there an explicit missing modality experiment (e.g., explicitly corrupt, mask out, or simply do not feed modality input into the model)? This is one of the claimed benefits in the paper, but it is unclear which tables show this (handling missing modality input better at inference time).

**Strengths And Weaknesses:**

**Strengths**:

Soundness: This paper has a clear motivation for learning multimodal clinical representations from small datasets with small models. Existing large models may overfit and rely on one modality, leading to suboptimal results. The method is presented clearly, with the standard knowledge distillation loss, plus unimodal classification losses for individual modalities, and finally, it distills knowledge from an ensemble of unimodal teachers. Balancing weights are introduced for the KD terms. Standard metrics such as AUROC and AUPRC on MIMIC-CXR, plus ablation studies on each component and results on other multimodal datasets are provided. The code is provided and seems cleanly written.

Audience: This paper provides a small network beneficial to clinical datasets and is useful for the medical ML community. The analysis of different KD loss components and the code can help the community.

**Weaknesses**:
See requested changes.

---

> ### Author Response · Authors · 2024-11-09
> **Authors’ Response to Review of Paper3419 by Reviewer sy5L**
>
> The authors would like to thank the reviewer for the thorough review and for highlighting relevant aspects. Below, we provide a point-by-point response to the questions raised in the requested changes, along with corresponding updates to the revised manuscript:
>
> > 1. Related work: Could the authors discuss [1], which is very similar to this work, where the knowledge is transferred by learning the teacher’s behavior within each modality, and weighting schemes are also introduced. The following similar related work seemingly lessens the novelty of this work.
>
> > [1] Jin, Woojeong, et al. "Msd: Saliency-aware knowledge distillation for multimodal understanding." arXiv preprint arXiv:2101.01881 (2021).
>
> Answer: We appreciate the comment of the reviewer. Certainly, Jin et al. (2021) has overlap with our work, such as proposing weighted modality-specific knowledge distillation. However, there are significant differences. The most relevant ones are described as follows:
>
> 1. Their work focuses on multimodal transformers, while our work is focused on multimodal joint fusion networks. The latter is a very popular choice  in healthcare settings due to the scale of medical datasets and transformers require very large datasets.  In this setting, an encoder is used per modality, and their outputs are fused (e.g., via concatenation or sum) and used as input for a linear layer, which outputs the final prediction. Therefore, in this context, the scope is significantly different (multimodal transformers vs. multimodal joint fusion networks).
>
> 2. We perform end-to-end training from scratch. No pre-training is performed on the student network. Both their teacher and student models are pre-trained models (i.e., fine-tuned BERT models). Therefore, the training procedure and model assumptions are significantly different.
>
> 3. We evaluate the performance of our proposed framework using different fusion methods. Their transformer-based approach does not involve fusion methods. In addition, we compare the performance of our work with strong baselines (i.e., end-to-end unimodal and multimodal models). However, they do not compare their approach with competitive baselines nor is it clear whether the ‘small model’ reported is fine-tuned or not, thus the benefits of their approach are unclear.
>
> 4. They follow the traditional offline KD paradigm, distilling from a single teacher to a student network. In contrast, we propose an ensemble of teachers to improve performance. Our ablation study shows that ensemble learning helps to improve performance with respect to a single teacher offline distillation.
>
> 5. They propose constrained weighting parameters (i.e., in the range of [0, 1]) based on saliency scores. In contrast, we do not restrict the weights to a specific range, which allows them to be used to overcome modality overfitting (i.e., balance multimodal learning) as our experiments demonstrate. In addition, their emphasis is on proposing several constrained weighting techniques, whereas our approach moves away from proposing complex weighting strategies and proposes a simple yet effective approach, as we integrate the weights as training hyper-parameters.
>
> 6. We integrate supervision losses, so the modality encoders can be used independently, in case of unimodal samples. Their approach does not consider this scenario, nor involves any added supervised loss allowing the use of unimodal encoders independently.
>
> 7. We open source our code, which can be readily used and reproduce our results and incorporate our approach to any joint fusion network. They do not provide any open source code to evaluate or reproduce their results. Their approach can only be integrated in similar transformer-based architectures.
>
> 8. In their approach, they use both teacher and student models with pre-trained checkpoints on models trained with huge NLP and vision datasets (even though the specifics about the specific checkpoint are not disclosed, the VisualBERT and TinyBert papers use COCO (328k samples) and large NLP datasets. Therefore, they use powerful pre-trained VisualBERT models as teachers, and powerful pre-trained TinyBERT as students, from the MMF library. These are both already distilled models from the base BERT model. For instance, in [2] it is specified that ‘*by performing the Transformer distillation on the text from the general domain, we obtain a general TinyBERT which provides a good initialization for the task-specific distillation*’. Therefore, despite not disclosing the particularities about this distillation (i.e., general or task-specific) this can make a huge difference in the student performance. Our student nor teacher models are not pre trained or previously distilled, they are all randomly initialized and trained from scratch.
>
> Based on all points above, we believe that our approach and [1] are significantly different in essential aspects such as scope, training procedure, model assumptions, and other methodological nuances.

---

> ### Author Response · Authors · 2024-11-09
> **Authors’ Response to Review of Paper3419 by Reviewer sy5L**
>
> > 2. Claim on missing modality: Could the authors clarify the first paragraph in 5.1: “while enabling unimodal predictions when modalities are missing”. Experiments in Table 1 show that unimodal distillation helps but did Table 1 explicitly test the missing modality scenario? In general for the paper, the reviewer apologizes in advance if she or he missed it, but is there an explicit missing modality experiment (e.g., explicitly corrupt, mask out, or simply do not feed modality input into the model)? This is one of the claimed benefits in the paper, but it is unclear which tables show this (handling missing modality input better at inference time).
>
> Answer: As the reviewer points out, Table 1 shows the multimodal performance of the models and does not consider any missing modality scenario. However, Table 2 and Table 3 evaluate the scenarios in which given that only one modality is present, the specific modality encoder can be leveraged to output a prediction for the unimodal sample (i.e., missing modality scenario). As mentioned above, this fact is distinctive from our approach as opposed to [1], as in the case of missing modality, without the need of masking, the modality encoders can be effectively leveraged to predict for unimodal samples. To further clarify this point, we have modified the abstract as follows: ‘*MIND involves multi-head joint fusion models, compared to single-head models, thereby enabling the utilization of the unimodal encoders in case of unimodal samples,* **without requiring imputation or masking**’. This text in bold substitutes the older version, which incorporated the ‘missing modality’ claim, making explicit our meaning. We have also modified related claims throughout the text to clarify their meaning.
>
> > 3. Broader Impact Concerns: A broader impact statement on societal issues, such as the potential risks of malicious usage of medical models, as well as implicit biases learned from knowledge distillation, should be added.
>
> Answer: We agree with the reviewer that these are important concerns. Following the reviewer suggestion, we have added a new section for the Broader Impact Statement that incorporates these relevant points (Section 7, before the conclusions). The whole statement reads as follows:
>
> **Broader Impact Statement**
>
> **Potential Risks of Malicious Usage of Medical Models**. *The deployment of machine learning models in healthcare carries significant risks if misused. Malicious actors could potentially exploit these models to manipulate medical diagnoses, treatment plans, or patient data. For instance, altering a model’s predictions could lead to incorrect diagnoses or inappropriate treatments, posing severe health risks to patients. In addition, the unauthorized access and manipulation of sensitive clinical data could lead to privacy breaches and identity theft.*
>
> **Implicit Biases Learned from Knowledge Distillation**. *Machine learning models trained on clinical datasets are susceptible to inheriting and amplifying existing biases present in the training data. Knowledge distillation, a model training process where a smaller model learns from larger, pre-trained models, can propagate these biases. This can result in unequal treatment outcomes for different demographic groups, exacerbating health disparities. For instance, if the training data contains biases against certain ethnicities or genders, the distilled model may continue to exhibit these biases, leading to unfair treatment recommendations.*
>
> **Privacy Concerns with MIMIC-IV and MIMIC-CXR data**. *The clinical datasets used in this research are MIMIC-IV (Johnson et al., 2023) and MIMIC-CXR (Johnson et al., 2019b). The authors would like to note that there are no privacy concerns related to these datasets. In particular, the MIMIC-IV and MIMIC-CXR datasets are de-identified and adhere to strict privacy regulations, ensuring that patient confidentiality is maintained (Johnson et al., 2024; 2019a). This allows researchers to utilize real-world clinical data for developing and testing machine learning models without compromising patient privacy.*
>
> *While machine learning holds great promise for advancing healthcare delivery, it is critical to consider and address these broader impact concerns. Ensuring robust security measures, mitigating biases, and maintaining patient privacy are essential steps towards the responsible and ethical use of machine learning in healthcare.*
>
> New references added to the revised version:
>
> Johnson, A. et al. (2024). MIMIC-IV (version 3.1). PhysioNet. https://doi.org/10.13026/kpb9-mt58.
>
> Johnson, A. et al. (2019a). MIMIC-CXR Database (version 2.0.0). PhysioNet. https://doi.org/10.13026/C2JT1Q.

---

> > ### Comment · Reviewer_sy5L · 2024-11-16
> > **Thank you for the response**
> >
> > The reviewer truly appreciates the authors' responses. All concerns have been addressed. The reviewer recommends acceptance of this paper.

---

### Review · Reviewer_djCo · 2024-11-01

**Summary Of Contributions:**

In this paper, the authors propose a multimodal model compression framework based on knowledge distillation that transfers knowledge from ensembles of pre-trained deep neural networks of varying sizes into smaller multimodal students. They evaluate the proposed method on binary classification and multilabel clinical prediction tasks using clinical time series data and chest X-ray images extracted from publicly available datasets.  The experimental results demonstrate that their method improves the performance of the smaller multimodal network across all five tasks, as well as fusion methods and multimodal network architectures, with respect to several state-of-the-art baselines.

**Audience:**

Yes

**Broader Impact Concerns:**

The authors claim their training data comes from public datasets. I think they might still need to explain that there are no privacy concerns when using those data. For example, they could include evidence of no privacy issues from the related dataset papers.

**Claims And Evidence:**

Yes

**Requested Changes:**

The authors are requested to make changes to address the weaknesses I mentioned.

**Strengths And Weaknesses:**

### **Strengths**
- This paper is well-organized and easy to read.
- The proposed method is technically sound.
- Extensive experimental results are provided to show the effectiveness of the proposed method.
- Detailed visualizations and ablation results are also included.

&nbsp;

### **Weaknesses**
- This paper mainly provides the experimental results on clinical prediction tasks. However, the proposed method can also be applied to conventional image classification or segmentation tasks. It would be interesting if the authors could also include the results on the conventional image tasks, e.g., on ImageNet or MS COCO.
- There are some typos and style inconsistencies in the paper. For example, "modality-specific" appears inconsistently, sometimes with a hyphen and sometimes without (e.g., "modality specific"). Phrases such as "modality utilization during multimodal training training" appear, with the word "training" repeated unnecessarily (in the caption of Figure 1).
- Some figures are very difficult to read. For example, there are too many lines in Figure 3. It is very difficult to compare different settings. It would be better to smooth the lines and use other strategies to make them easier to understand.
- The technical contributions seem incremental. Seeing from Figure 1, it looks like the proposed method is very similar to Hayat et al. (2022). The only difference between A2 and the proposed method is using weighting hyper-parameters. Even though the novelty is not a mandatory requirement according to the TMLR reviewing guidelines, the authors still need to adjust their claims if the proposed method is very similar to Hayat et al. (2022).

---

> ### Author Response · Authors · 2024-11-09
> **Authors’ Response to Review of Paper3419 by Reviewer djCo**
>
> The authors would like to thank the reviewer for the thorough review and for highlighting relevant aspects. Below, we provide a point-by-point response to the questions raised in the requested changes, along with corresponding updates to the revised manuscript:
>
> > 1. This paper mainly provides the experimental results on clinical prediction tasks. However, the proposed method can also be applied to conventional image classification or segmentation tasks. It would be interesting if the authors could also include the results on the conventional image tasks, e.g., on ImageNet or MS COCO.
>
> Answer: We appreciate the comment of the reviewer. The MIND framework is designed to work with multimodal datasets, that is, incorporating more than one data modality (e.g., image and audio). The ImageNet and MS COCO datasets are primarily image datasets, widely used as benchmarks for a wide variety of computer vision tasks. Addressing the comment of the reviewer, in order to assess the generalization of our framework beyond clinical tasks, we refer the reviewer to Section 5.5 (Additional Results on Multimodal Benchmark Datasets), where we provide an extensive validation of the proposed methodology on non-clinical benchmark multimodal datasets and tasks. In particular, we evaluate the proposed method and SOTA baselines on the CREMA-D dataset (audio-visual dataset for emotion recognition), S-MNIST (audio-visual dataset for digit classification), and LUMA (audio-visual dataset for object recognition).
>
> > 2. There are some typos and style inconsistencies in the paper. For example, "modality-specific" appears inconsistently, sometimes with a hyphen and sometimes without (e.g., "modality specific"). Phrases such as "modality utilization during multimodal training training" appear, with the word "training" repeated unnecessarily (in the caption of Figure 1).
>
> Answer: We appreciate the attention to detail of the reviewer. We apologize for the typos and style inconsistencies found in the original version of the manuscript. We have addressed these issues and others, performing a complete proofreading of the entire manuscript for the revised version. We have highlighted the changes and corrections in blue.
>
> > 3. Some figures are very difficult to read. For example, there are too many lines in Figure 3. It is very difficult to compare different settings. It would be better to smooth the lines and use other strategies to make them easier to understand.
>
> Answer: We agree that the original Figure 3 was a bit complicated to understand as the lines were too cluttered. Following the reviewer comment, we have smoothened out the lines (i.e., plotting the values every 5 epochs instead of every epoch) and stretched out the vertical axes to create more separation between the lines. We believe that this addresses the problem and allows for a better understanding of the graph. We appreciate the reviewer's help in making this graph easier to understand. We have included the new figure in the update version of the manuscript (page 11).

---

> ### Author Response · Authors · 2024-11-09
> **Authors’ Response to Review of Paper3419 by Reviewer djCo**
>
> > 4. The technical contributions seem incremental. Seeing from Figure 1, it looks like the proposed method is very similar to Hayat et al. (2022). The only difference between A2 and the proposed method is using weighting hyper-parameters. Even though the novelty is not a mandatory requirement according to the TMLR reviewing guidelines, the authors still need to adjust their claims if the proposed method is very similar to Hayat et al. (2022).
>
> Answer: We appreciate the reviewer’s comment. Hayat et al. (2022), illustrated in Figure 1.A.1, serves as a classical example of a joint fusion multimodal network, where outputs from unimodal encoders are fused and processed by an additional network to produce the final multimodal prediction. Our research focuses on joint fusion multimodal networks, particularly in clinical applications, which is why we selected Hayat et al. (2022) as the state-of-the-art (SOTA) backbone for our implementation. However, any other SOTA clinical fusion model could also be utilized as a backbone.
>
> While Hayat et al. (2022) is a SOTA clinical multimodal fusion network, there are significant differences between our work and theirs. Key distinctions include:
> - **Modality-Specific Classification Heads**: Our framework includes modality-specific classification heads (Figure 1.A.2), which are a prerequisite of our framework and not part of Hayat et al. (2022).
> - **MIND Framework Integration**: We implement teacher ensembling and modality weighting to fully incorporate the MIND framework. These features are absent in Hayat et al. (2022).
> - **Training Objective (loss)**: Our basic loss for two modalities incorporates five components: three supervision losses (one multimodal and two modality-specific losses) and two weighted modality-specific knowledge distillation losses. In contrast, Hayat et al. (2022), like other multimodal joint fusion networks, has an objective function with a single component, namely, a multimodal supervision loss.
> - **Generalization Assessment**: We assess the generalization of our framework using various multimodal architectures and fusion methods, distinct from those used in Hayat et al. (2022).
> - **Extensive Validation**: Our framework includes extensive validation on different joint fusion multimodal networks, as detailed in Section 5.5, which is not covered in Hayat et al. (2022).
>
> Our proposal is designed to be adaptable to any joint fusion multimodal network, with Hayat et al. (2022) serving as SOTA backbone for our clinical focus, which is the main theme of our paper. We demonstrate this adaptability by assessing the generalization of our framework, employing additional multimodal architectures and fusion methods described in Appendix C.2, which are distinct from those used in Hayat et al. (2022). In summary, Figure 1.A.1 depicts MedFuse (Hayat et al. (2022)), which does not involve modifying the backbone framework to include modality-specific classification heads or the integration of teacher ensembling and modality weighting.
>
> > 5. Broader Impact Concerns: The authors claim their training data comes from public datasets. I think they might still need to explain that there are no privacy concerns when using those data. For example, they could include evidence of no privacy issues from the related dataset papers.
>
> Answer: We agree with the reviewer that this is an important concern. Following the reviewer's suggestion, we have added this relevant point to the new Broader Impact Statement (Section 7). In particular, this part reads as follows:
>
> **Privacy Concerns with MIMIC-IV and MIMIC-CXR data**. *The clinical datasets used in this research are MIMIC-IV (Johnson et al., 2023) and MIMIC-CXR (Johnson et al., 2019b). The authors would like to note that there are no privacy concerns related to these datasets. In particular, the MIMIC-IV and MIMIC-CXR datasets are de-identified and adhere to strict privacy regulations, ensuring that patient confidentiality is maintained (Johnson et al., 2024; 2019a). This allows researchers to utilize real-world clinical data for developing and testing machine learning models without compromising patient privacy.*
>
> New references added to the revised version:
>
> Johnson, A. et al. (2024). MIMIC-IV (version 3.1). PhysioNet. https://doi.org/10.13026/kpb9-mt58.
>
> Johnson, A. et al. (2019a). MIMIC-CXR Database (version 2.0.0). PhysioNet. https://doi.org/10.13026/C2JT1Q.

---

### Decision · Action_Editor_ot7o · 2024-12-09

**Recommendation:** Accept as is

**Comment:**

In this paper, the authors presented a knowledge distillation framework for multimodal clinical prediction tasks. Specifically, a teacher-student framework was proposed, where knowledge was transferred from ensembles of pre-trained deep neural networks of varying sizes into smaller student models. Experimental results on a few datasets show the effectiveness of the proposed method.

Overall, this paper presented some interesting findings, which were appreciated by the reviewers, receiving all positive recommendations (2 Leaning Accept, and 1 Accept). However, there are concerns about the novelty and significance of the proposed method. The technical contributions were found to be incremental (details please see the review comments), when compared to a few related works. But considering the motivation, technical soundness, the validated claims, and the findings this paper has presented, it is considered to be of interest to the audience of TMLR. Therefore, the AE recommends Accept.

**Audience:**

There would be a group of people in TMLR's audience who would be interested in knowing the findings of this paper.

**Claims And Evidence:**

The claims made in the submission are supported by accurate, convincing and clear evidence.